# A chemical biology screen identifies a vulnerability of neuroendocrine cancer cells to SQLE inhibition

Christopher E. Mahoney [1], David Pirman[1], Victor Chubukov[1], Taryn Sleger[1], Sebastian Hayes [1], Zi Peng Fan[1], Eric L. Allen[1], Ying Chen[2], Lingling Huang[2], Meina Liu[2], Yingjia Zhang[2], Gabrielle McDonald[1], Rohini Narayanaswamy[1], Sung Choe[1], Yue Chen[1], Stefan Gross [1], Giovanni Cianchetta[1], Anil K. Padyana[1], Stuart Murray[1], Wei Liu[1], Kevin M. Marks[1], Joshua Murtie[1], Marion Dorsch[1], Shengfang Jin[1], Nelamangala Nagaraja[1], Scott A. Biller[1], Thomas Roddy[1], Janeta Popovici-Muller[1,3] & Gromoslaw A. Smolen [1,4]

Aberrant metabolism of cancer cells is well appreciated, but the identification of cancer subsets with specific metabolic vulnerabilities remains challenging. We conducted a chemical biology screen and identified a subset of neuroendocrine tumors displaying a striking pattern of sensitivity to inhibition of the cholesterol biosynthetic pathway enzyme squalene epoxidase (SQLE). Using a variety of orthogonal approaches, we demonstrate that sensitivity to SQLE inhibition results not from cholesterol biosynthesis pathway inhibition, but rather surprisingly from the specific and toxic accumulation of the SQLE substrate, squalene. These findings highlight SQLE as a potential therapeutic target in a subset of neuroendocrine tumors, particularly small cell lung cancers.

[1] Agios Pharmaceuticals, 88 Sidney Street, Cambridge, MA 02139, USA. [2] Shanghai ChemPartner Co. Ltd., 998 Halei Road, Pudong, 201203 Shanghai, China. [3]Present address: Decibel Therapeutics, 1325 Boylston Street, Suite 500, Boston, MA 02215, USA. [4]Present address: Celsius Therapeutics, 215 First Street, Cambridge, MA 02142, USA. Correspondence and requests for materials should be addressed to G.A.S. (email: gsmolen@celsiustx.com)

The concept of precision cancer medicine, wherein tumor genotype guides the selection of appropriate targeted therapies, has transformed the clinical practice of cancer treatment. Multiple targeted agents have shown dramatic results in specific, genetically defined subpopulations, such as epidermal growth factor receptor (EGFR) inhibitors in EGFR-mutant lung tumors and BRAF inhibitors in BRAF-mutant melanomas[1]. Unfortunately, relatively few patients harbor clinically actionable mutations[2], suggesting that alternative approaches, such as expanding the scope of drugging strategies and alternative patient selection criteria, will be needed to address the majority of cancer cases

Screening cancer cell lines for sensitivity to small molecules has emerged as a powerful tool to identify context-specific vulnerabilities. The approach is scalable and some recent studies have assessed hundreds of cell lines for their sensitivity to hundreds of small molecules[3–5]. While the screens can be limited by the diversity of the cell lines, small molecules, and the specifics of the assay used, the unbiased nature of such screens allows for de novo hypothesis generation, particularly when coupled with increasingly deeper characterization of the cell lines utilized. While early screens focused on drug sensitivities driven by single tumor-associated mutations, the latest efforts have highlighted growth sensitivities driven by multi-parametric biomarker signatures[6] or differentiation-based vulnerabilities associated with lineage[7], clearly illustrating the advantages of the continued expansion of screening formats and analytical capabilities.

Here we report a chemical biology screen in hundreds of cancer cell lines leading to the identification of a subset of neuroendocrine cell lines, particularly within the small cell lung cancer (SCLC) lineage, that displays a remarkable sensitivity to NB-598. NB-598 is a known inhibitor of squalene epoxidase (SQLE), an enzyme in the cholesterol biosynthetic pathway catalyzing the conversion of squalene to 2,3-oxidosqualene[8]. Using several independent pharmacological and genetic approaches, we demonstrate that the cellular effects of NB-598 are on target and appear to be related to the accumulation of squalene, a substrate of the SQLE enzyme. SQLE sensitivity is unique, as inhibition of other steps in the cholesterol biosynthetic pathway does not recapitulate the same pattern of sensitivity in SCLC cell lines. Our findings support further investigation of SQLE as a therapeutic target in a distinct subset of SCLC.

## Results

**SCLC cell lines display sensitivity to NB-598.** To identify novel cancer vulnerabilities, we screened a panel of 482 cell lines with a diverse set of metabolic inhibitors. NB-598, an SQLE inhibitor[8], displayed fairly specific activity in a subset of cell lines, particularly in neuroblastoma and lung cancer cell lines (Fig. 1a and Supplementary Data 1). Analysis of expression patterns in sensitive cell lines revealed enrichment of multiple gene ontology (GO) biological processes linked to neurogenesis and neural development (Fig. 1b). Given that SCLC is thought to arise from neuroendocrine cells in the lung[9], we tested the NB-598 sensitivity in a panel of 42 SCLC cell lines. We calculated a quantitative metric of sensitivity for each cell line based on the area under the curve (AUC) of the mu/mu.max curve to more accurately capture the potency and extent of NB-598 effects. Interestingly, the degree of NB-598 sensitivity was highly varied, with cell death evident in some cell lines (mu/mu.max < 0). We categorized the SCLC cell lines as sensitive (5/42), moderate (11/42), and insensitive (26/42) (Fig. 1c and Supplementary Data 2) and focused all subsequent efforts on this indication. Analysis of genetic mutations and copy number alterations in SCLC cell lines did not yield any associations with NB-598 sensitivity (data not

shown). To further understand the patterns of sensitivity, we conducted RNA sequencing (RNA-Seq; Supplementary Data 3) and proteomic (Supplementary Data 4) characterization of the SCLC panel to identify unbiased expression signatures associated with enhanced NB-598 response (Supplementary Fig. 1 and Supplementary Fig. 2). Given the growing understanding that SCLC tumors can be further subdivided based on the status of lineage-defining transcription factors[10], ASLC1 and NEUROD1, we specifically investigated NB-598 response as the function of ASCL1 and NEUROD1 expression levels. Interestingly, we noticed a pattern where all of the sensitive cell lines displayed high levels of ASCL1 and low levels of NEUROD1 (Fig. 1d). To further dissect the NB-598 response within the ASCL1-high/NEUROD-low subset of 25 cell lines, we identified RNA-Seq and proteomics expression signatures associated with NB-598 sensitivity (Supplementary Fig. 3 and Supplementary Fig. 4). Several commonly used neuroendocrine markers, such as CHGA and SYP, were found to be expressed at significantly higher levels in NB-598-sensitive cell lines (Supplementary Fig. 5a) but were not part of the expression signatures identified (Supplementary Figs. 1–4) due to the stringent cutoffs employed. The robustness of these expression signatures to prospectively identify cell lines sensitive to NB-598 will have to be tested in future studies utilizing larger panels of cell lines.

For all the subsequent functional analyses, we assembled a panel of representative cell lines: ASCL1-high/NEUROD1-low cell lines that are NB-598 sensitive (LU139, H146, H1963) and ASCL1-low/NEUROD1-high cell lines that are NB-598 insensitive (H82, H2171, H446). Since not all ASCL1-high/NEUROD1-low cell lines are sensitive to NB-598, we also included an insensitive cell line (H2081) from this category to more accurately represent the diversity of SCLC cell lines. Expected expression levels of ASCL1 and NEUROD1 were confirmed by quantitative PCR (qPCR; Supplementary Fig. 5b, c).

**NB-598 specifically inhibits SQLE.** To confirm that NB-598 inhibits SQLE in SCLC cells, we interrogated the cholesterol biosynthetic pathway using several different assays. First, we analyzed the NB-598 effects on de novo cholesterol biosynthesis, which can be monitored by exposing cells to $^{13}C_2$-labeled acetate. Co-treatment of labeled acetate with 1 µM NB-598 resulted in dramatic suppression of the entire population of labeled isotopomers of cholesterol (Fig. 2a). Given the highest degree of labeling in the cholesterol m+2 isotopomer, we used it as an assay readout to measure the extent of SQLE inhibition. NB-598 potently suppressed de novo cholesterol biosynthesis with a measured IC50 of 76 nM (Fig. 2b). To further confirm the specificity of NB-598 effects, we took advantage of the recently developed structural analog of NB-598 that is devoid of biochemical activity, called NB-598.ia[11]. As expected, NB-598.ia did not show any effects on de novo cholesterol biosynthesis (Fig. 2b). Second, we monitored the levels of squalene, since the inhibition of SQLE is expected to result in the accumulation of the direct substrate of the SQLE enzymatic reaction. As predicted, treatment of cancer cells using NB-598 resulted in a dramatic accumulation of squalene in a dose-dependent fashion (Fig. 2c). Finally, we directly compared NB-598 and NB-598.ia compounds in a small panel of representative SCLC cell lines and showed that, in contrast to NB-598, NB-598.ia did not inhibit cellular growth (Fig. 2d). Collectively, this set of highly related but functionally distinct pharmacological tools suggests that observed NB-598 effects are likely to result from specific SQLE inhibition.

**In vivo efficacy of NB-598.** To investigate the in vivo SQLE dependence of the identified SCLC cell lines, we first conducted

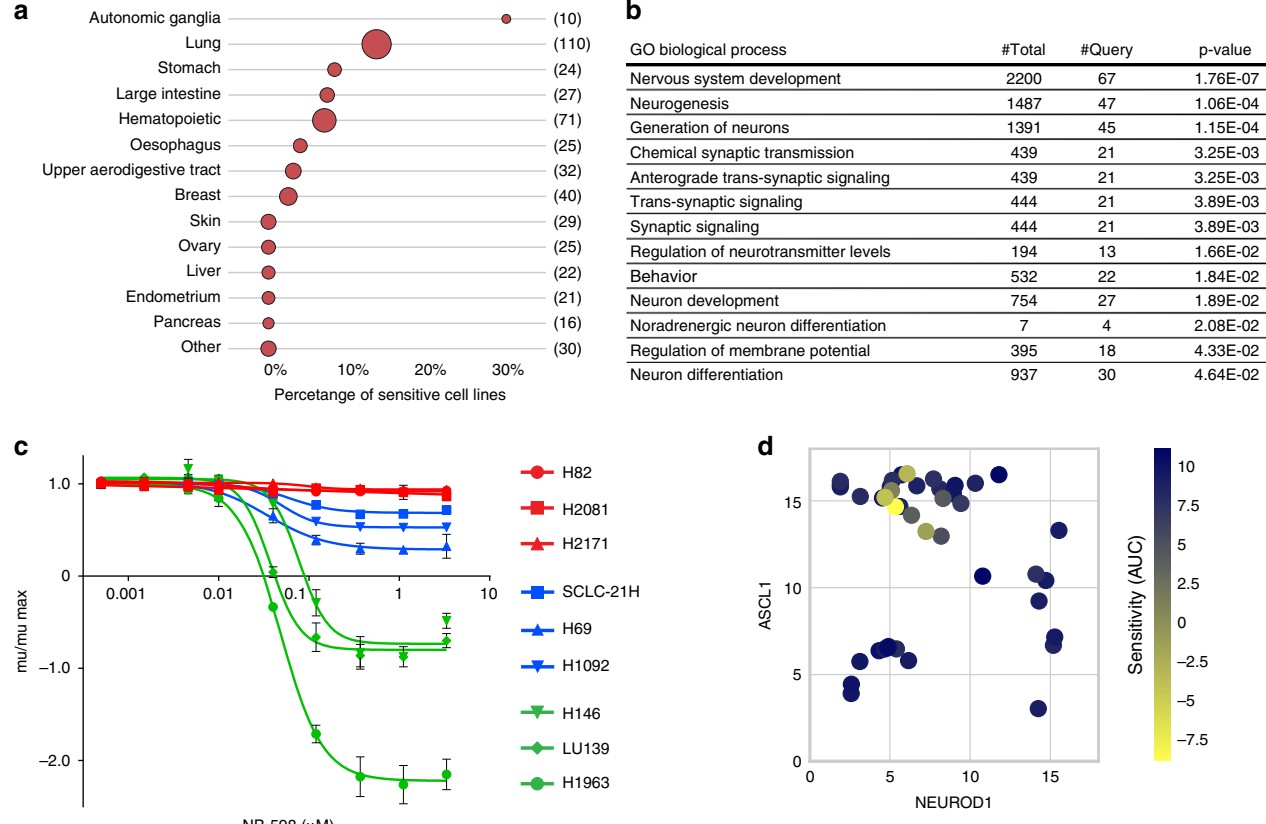

**Fig. 1** A subset of SCLC cancer cell lines is sensitive to NB-598. **a** Summary of NB-598 sensitivities observed in a panel of 482 cancer cell lines. Number in parentheses represents the total number of cell lines within a particular category and the size of the circle is proportional to the number of sensitive cell lines within each category. Cells were defined as sensitive based on GI75 < 2 µM. **b** Gene ontology enrichment of gene expression in cell lines sensitive to NB-598. Gene expression data from the CCLE database was obtained for 451 cell lines and genes differentially expressed in cells sensitive to NB-598 (t test, p < 0.001) were analyzed for statistically overrepresented GO categories. GO categories were restricted to those with <3000 genes. **c** NB-598 sensitivity in a representative panel of SCLC cell lines. Cell lines were categorized based on the measured AUC of mu/mu.max curve and were defined as sensitive (AUC < 2), moderate (2 < AUC < 9), and insensitive (AUC > 9). Classification categories are shown in green, blue, and red colors, respectively. Mu/mu.max calculations were used to compare growth rates of drug-treated to DMSO-treated cells (mu.drug/mu.DMSO), where maximum growth is observed in DMSO-treated cells. Value of mu/mu.max = 1 corresponds to no effect of drug added. Values of mu/mu.max < 0 denote cytotoxic effects (fewer cells at Tend than T0). Values of mu/mu.max between 0 and 1 denote various extents of partial growth inhibition. Mean values of triplicate measurements from a representative experiment are plotted and error bars represent s.d. **d** Relative mRNA expression levels of *ASCL1* and *NEUROD1* as a function of NB-598 sensitivity. Expression levels are represented as log2-transformed normalized counts. AUC-based NB-598 sensitivity of individual SCLC cell lines is indicated on a blue-to-yellow scale

an assessment of NB-598 behavior by monitoring compound concentration in plasma and tumors, as well as the corresponding squalene accumulation in the tumors. We administered a single dose of NB-598 at 300 mg kg$^{-1}$ to mice bearing H146 xenografts and observed NB-598 half-life in circulating plasma to be approximately 8.35 h (Fig. 3a). Interestingly, NB-598 appeared to accumulate and persist in the H146 tumors as long as 48 h after the initial dose, perhaps due to the very lipophilic nature of this compound. As expected, squalene accumulation was rapidly induced in the tumors following NB-598 dosing (Fig. 3b). These results suggested that once-daily NB-598 dosing would likely be sufficient to effectively inhibit SQLE in human xenograft models.

To determine the extent of SQLE inhibition in vivo, we conservatively treated tumor-bearing mice with three daily doses of NB-598 to ensure that a steady-state level of inhibitor was reached and we subsequently measured tumor-specific de novo cholesterol biosynthesis inhibition using D$_2$O labeling. D$_2$O labeling has the advantage of rapid equilibration within the body and provides a robust signal as multiple deuterium atoms can get

incorporated into cholesterol[12,13]. Using increasing doses of NB-598, we showed that 300 mg kg$^{-1}$ dose results in almost complete inhibition of de novo cholesterol biosynthesis (Fig. 3c).

Using the 300 mg kg$^{-1}$ NB-598 dose, we conducted a series of in vivo efficacy studies in several cell lines that were either sensitive or insensitive to NB-598 in vitro. To determine the role of SQLE in the context of tumor maintenance, we started the dosing only after tumors were well established, at approximately 200 mm$^3$. We found excellent translation of the in vitro observations to in vivo results, wherein 3/3 SQLE-sensitive cell lines displayed profound sensitivity to NB-598 and 2/2 SQLE-insensitive cell lines showed no significant effect (Fig. 3d). Repeated doses of NB-598 were well tolerated throughout the course of the studies, with no overt signs of toxicity (data not shown). We also collected H1963 tumors at the end of the efficacy study and assessed several different markers to understand the mechanisms of NB-598 action in vivo. We observed no major differences in staining with antibodies against Ki-67 and against cleaved caspase 3, representing diagnostic markers for

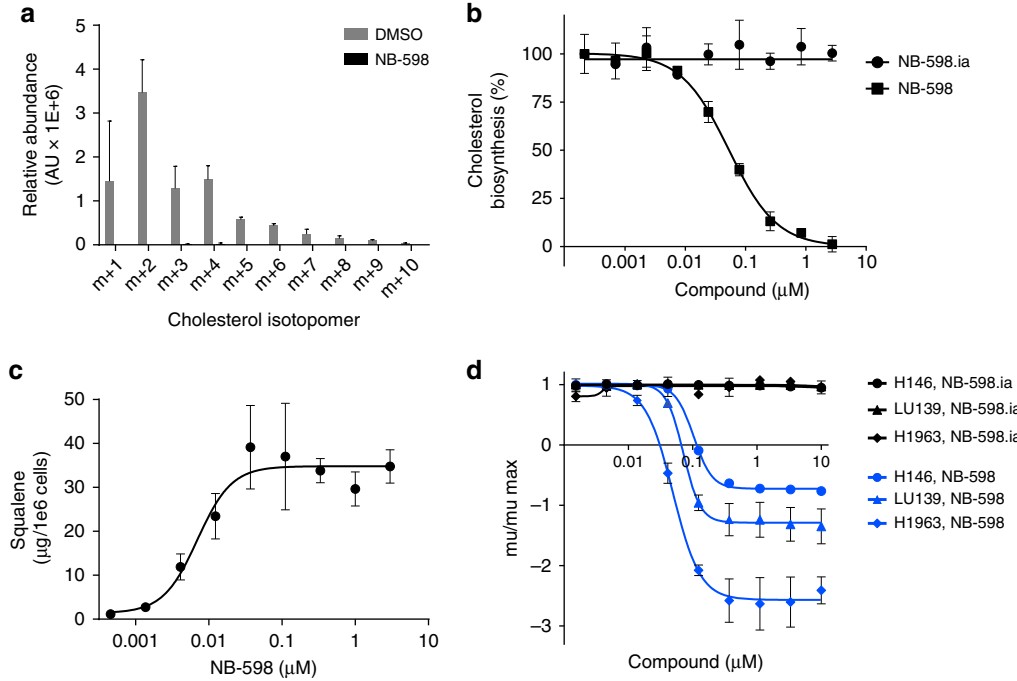

**Fig. 2** NB-598 is a specific SQLE inhibitor. **a** Cholesterol isotopomer distribution in a $^{13}C_2$-acetate labeling experiment. H1963 cells were co-treated with 1 μM NB-598 and 500 μM $^{13}C_2$-acetate for 48 h. **b** De novo cholesterol biosynthesis inhibition measured in H1963 cells using $^{13}C_2$-acetate labeling. Effects on cholesterol m+2 isotopomer were quantitated to generate the dose–response curves. **c** Squalene accumulation measured in LU139 cells after 24 h of NB-598 treatment. **d** NB-598 and NB-598.ia sensitivity in a representative panel of SQLE-sensitive cell lines. For all panels, mean values of triplicate measurements from a representative experiment are plotted and error bars represent s.d

proliferation and apoptosis, respectively (Supplementary Fig. 6a, b). It is important to note that these studies utilized differently sized tumors at the end of the study, where approximately 85% tumor growth inhibition was observed, and thus more careful time-course experiments will be needed to fully dissect the biological mechanisms involved.

Collectively, these results suggest that the NB-598 sensitivity patterns observed in vitro appear to be recapitulated in vivo and thus highlight the specificity and robustness of NB-598 effects.

**The cholesterol pathway and NB-598 sensitivity**. To elucidate the differences between SQLE-sensitive and SQLE-insensitive cell lines, we assessed cholesterol biosynthetic pathway status and regulation in SCLC cell lines. Because the cholesterol biosynthetic pathway is exquisitely controlled by multiple regulatory mechanisms[14], we investigated the baseline levels of expression of several key proteins in the pathway. While we observed some trends, such as somewhat higher levels of HMGCR in SQLE-sensitive cells, there were no dramatic differences in the protein levels of LDLR, FDFT1, and SQLE (Fig. 4a and Supplementary Data 4). To functionally measure cholesterol biosynthetic flux in cell lines, we used $D_2O$ labeling and measured the incorporation of deuterium into cholesterol. This method bypasses the caveats associated with $^{13}C_2$-acetate labeling and the possible differential uptake of the labeled metabolite among the cell lines. While we observed some differences among cell lines, the baseline level of pathway flux was surprisingly similar between the SQLE-sensitive and SQLE-insensitive cell lines (Fig. 4b).

We next hypothesized that the differential response to NB-598 might underlie the dramatically different cellular responses observed in the panel of SCLC cell lines. To ensure focus on the early part of the response and not the downstream secondary effects, we undertook a temporal dissection of the response to

NB-598 in vitro. Growth defects observed in multiple SQLE-sensitive cell lines appeared to be due to the induction of apoptosis, which was particularly apparent 72 h after the addition of NB-598 addition (Fig. 4c). Based on these results, we focused on characterizing responses to NB-598 at a point prior to widespread apoptosis in SQLE-sensitive cells. First, we measured squalene accumulation in a representative panel of cell lines and showed, surprisingly, that cellular response to NB-598 does not correlate with the amounts of accumulated squalene (Fig. 4d). To determine whether squalene secretion out of the cells might account for differences between SQLE-sensitive and SQLE-insensitive cell lines, we measured squalene in media during the course of NB-598 treatment. No major differences were observed (Fig. 4e), even in the cell lines displaying high squalene accumulation, suggesting that squalene secretion is unlikely to account for the dramatic differences in NB-598 responses.

Since the inhibition of cholesterol biosynthesis is well documented to result in coordinated upregulation of multiple biosynthetic steps via the action of SREBP2 transcription factor[15], we assessed the functional status of this feedback loop in multiple SCLC cell lines. Treatment of SCLC cell lines with 1 μM NB-598 for 24 h resulted in expected upregulation of multiple genes in the cholesterol pathway, such as *HMGCR*, *FDFT1*, *SQLE*, and *LDLR* (Fig. 4f). While somewhat stronger responses were observed in SQLE-sensitive cell lines, the differences were not compelling. Collectively, these data are consistent with the interpretation that cholesterol pathway status and its regulation are unlikely to be the sole determinants of sensitivity to SQLE inhibition in SCLC cell lines.

**Squalene accumulation is critical for NB-598 efficacy**. To further understand the mechanisms of sensitivity to SQLE inhibition, we used a CRISPR-based suppressor screen to identify genes

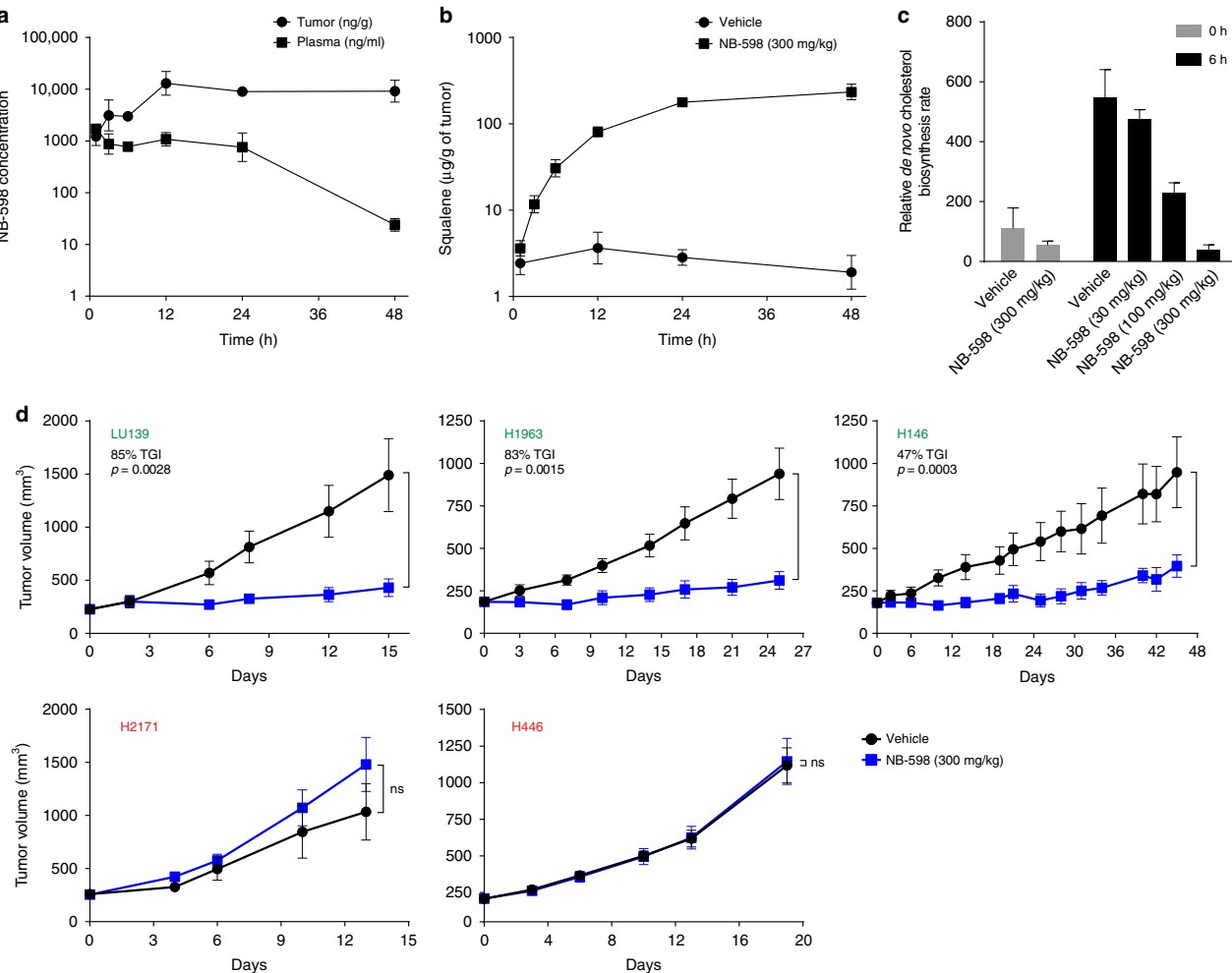

**Fig. 3** In vivo translation of SQLE sensitivity in a subset of SCLC cell lines. **a** NB-598 single-dose (300 mg kg$^{-1}$) pharmacokinetic profile in H146 xenograft-bearing mice. Mean values of triplicate measurements from a representative experiment are plotted and error bars represent s.e.m. **b** Squalene accumulation in H146 xenografts after a single dose of NB-598 (300 mg kg$^{-1}$). Mean values of triplicate measurements from a representative experiment are plotted and error bars represent s.e.m. **c** De novo cholesterol biosynthesis in tumors, as measured by deuterium enrichment in cholesterol after D$_2$O labeling. Mean values of triplicate measurements from a representative experiment are plotted and error bars represent s.d. **d** In vivo efficacy of NB-598 (300 mg kg$^{-1}$) dosed daily in multiple xenograft models. Dosing was initiated once tumors were approximately 200 mm$^3$. SQLE-sensitive and SQLE-insensitive cell lines are denoted in green and red, respectively. Mean values of each experimental group ($n = 10$–15 animals) from a representative experiment are plotted and error bars represent s.e.m. Student's $t$ test p values are shown and ns indicates $p > 0.05$

that, when inactivated, can lead to NB-598 resistance. Since SCLC cell lines are notoriously difficult to manipulate using lentiviral methods, we employed an adherent lung cancer line that also displayed NB-598 sensitivity, A427 (Supplementary Fig. 7a). We chose conditions of 50 nM NB-598 that result in complete cell killing during a 2–3-week course of treatment. The introduction of a custom CRISPR library targeting approximately 6000 genes covering metabolome, kinome, and epigenome conferred NB-598 resistance to a small subpopulation of cells at 3 weeks of treatment. We compared single-guide RNA (sgRNA) abundance before and after NB-598 treatment using median log2-fold change of the abundance of all sgRNAs targeting a given gene (Supplementary Data 5). Strikingly, we identified multiple genes in the cholesterol biosynthetic pathway that are upstream of the SQLE step, including *ACAT2*, *HMGCR*, and *FDFT1* (Fig. 5a). To confirm these findings, we focused on *FDFT1*, as the most proximal upstream gene to *SQLE*, and developed FDFT1 KO clones in A427 cells, as well as in another independent cell line, LK2, that displayed NB-598 sensitivity (Supplementary Fig. 7b). FDFT1 KO

clones displayed loss of full-length protein expression (Fig. 5b). Consistent with the screen results, FDFT1 KO cells in both cell lines resulted in dramatic rescue of NB-598 sensitivity (Fig. 5c). Similar results were obtained in several SCLC cell lines using a pharmacological FDFT1 inhibitor, lapaquistat, and an HMGCR inhibitor, atorvastatin (Fig. 5d and Supplementary Fig. 7c, d). As expected, co-treatment with NB-598 and lapaquistat resulted in a dramatic reduction of squalene accumulation, as compared to NB-598 treatment alone (Fig. 5e). To confirm these results using an orthogonal approach, we employed two independent sgRNAs targeting SQLE and showed that CRISPR-mediated KO in LK2 cells results in profound growth defects (Fig. 5f). Consistent with the previous observations using NB-598 to inhibit SQLE function, the growth defects were rescued by the knockout of the upstream enzyme, FDFT1.

To further demonstrate the unique nature of SQLE inhibition, we employed a panel of pharmacological inhibitors upstream and downstream of SQLE in a small panel of representative cell lines (Fig. 5g). Multiple inhibitors upstream of SQLE, such as

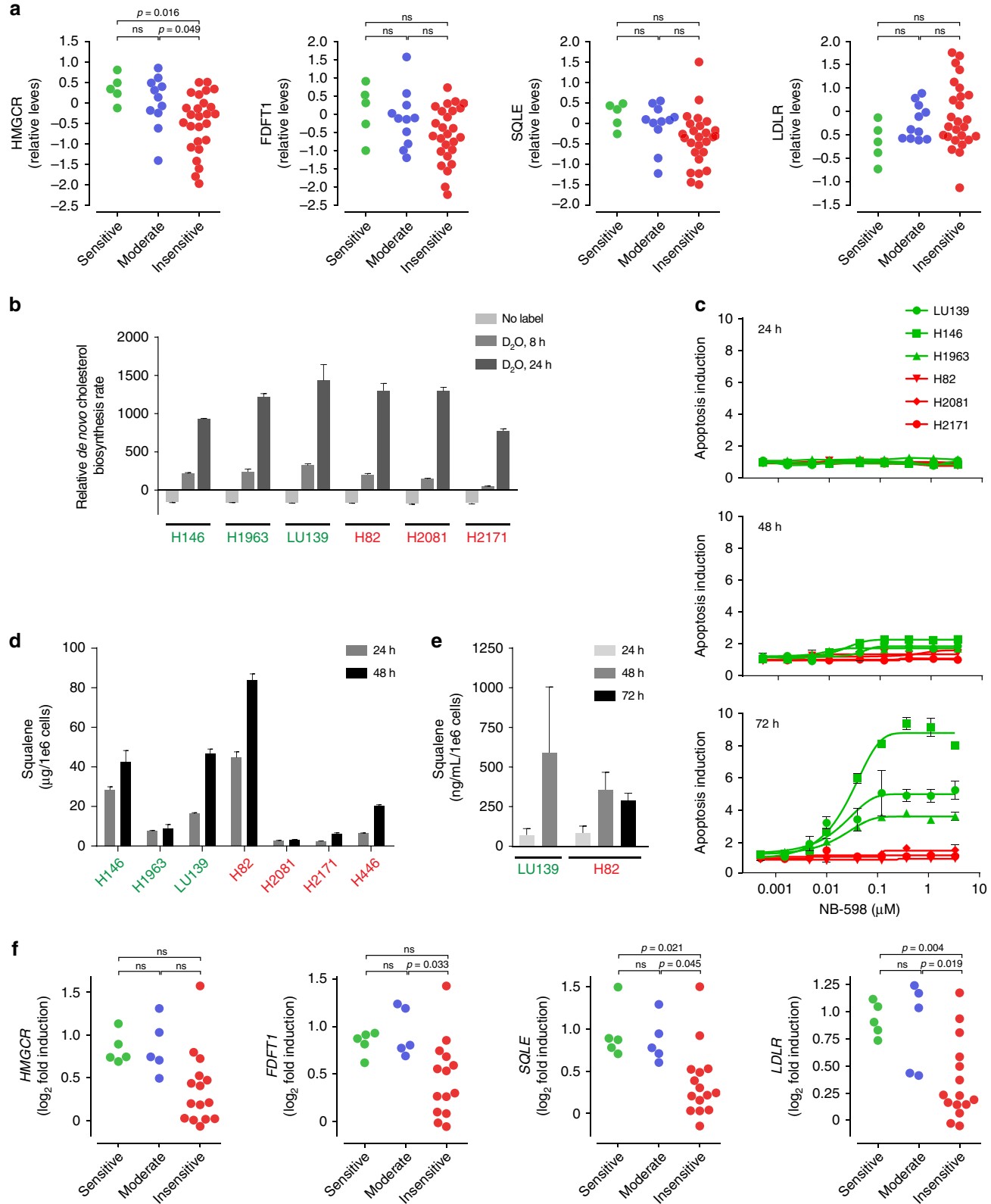

fluvastatin and atorvastin inhibiting HMGCR[16] and lapaquistat inhibiting squalene synthase[17], did not display a similar sensitivity profile to NB-598 (Fig. 5h). Similarly, agents downstream of SQLE, such as Ro48–8071 inhibiting lanosterol synthase[18], azalanstat inhibiting lanosterol C14-alpha demethylase[19], SR31747 inhibiting 7,8-sterol isomerase[20], and AY9944 inhibiting $\Delta^7$-sterol reductase[21,22], did not show significant sensitivity in the cell lines tested (Fig. 5h). Collectively, this data suggest that NB-598 sensitivity does not appear to represent a generic cholesterol pathway-level requirement and that accumulation of SQLE substrate, squalene, is critical for the observed growth inhibitory effects.

**Fig. 4** Cholesterol pathway status in SQLE-sensitive and SQLE-insensitive cell lines. **a** Baseline protein expression levels of key regulatory points of the cholesterol biosynthetic pathway, selected from the proteomics dataset. Student's *t* test *p* values are shown and ns indicates *p* > 0.05. **b** Baseline de novo cholesterol biosynthesis in cell lines, as measured by deuterium enrichment in cholesterol after $D_2O$ labeling. **c** Kinetics of cell death induction following NB-598 treatment assessed by caspase 3/7 activity. **d** Squalene accumulation in cell pellets in a panel of SCLC cell lines after 1 μM NB-598 treatment for 24 h. **e** Squalene accumulation in media from SCLC cell lines after 1 μM NB-598 treatment for 24 h. **f** Relative induction of cholesterol pathway genes, as measured by qPCR, after 1 μM NB-598 treatment for 24 h. DMSO treatment was used as a control. Student's *t* test *p* values are shown and ns indicates *p* > 0.05. For **b**–**e**, mean values of triplicate measurements from a representative experiment are plotted and error bars represent s.d. All cell lines are color coded based on NB-598 sensitivity, as described in Fig. 1

**Squalene storage and lipid droplets**. To further understand the role of squalene in exerting its effect on SCLC cell lines, we conducted a series of imaging experiments using transmission electron microscopy (TEM) in representative SQLE-sensitive and SQLE-insensitive cell lines at a relatively early time point, 24 h, before the onset of apoptosis. We noticed a striking accumulation of darkly stained vesicles in all NB-598-treated cells (Fig. 6a). These vesicles most likely represent lipid droplets containing squalene, since a similar pattern of staining was observed using BODIPY neutral lipid stain (Fig. 6b). No obvious differences were observed in accumulating lipid droplets between representative SQLE-sensitive and SQLE-insensitive cell lines (Fig. 6b).

To determine the role of lipid droplets, we abrogated their formation by inhibiting the synthesis of their two critical components, triacylglycerols and sterol esters, using pharmacological inhibitors of diacylglycerol *O*-acyltransferases (DGATs) and sterol *O*-acyltransferases (SOATs), respectively[23]. Treatment of DGAT and SOAT inhibitors was able to dramatically alter the pattern of BODIPY dye staining after NB-598 treatment, wherein cells no longer showed clear lipid droplets but rather diffuse lipid staining (Fig. 6c). Interestingly, in the context of abrogated lipid droplets, multiple SQLE-insensitive cell lines displayed sensitivity to NB-598 suggesting that lipid droplets can serve as protective squalene storage depots (Fig. 6d). These results imply that cellular squalene storage capacity might contribute to the differential NB-598 sensitivity observed among different SCLC cell lines.

To further test the relationship between squalene levels and differential NB-598 sensitivity, we utilized a well-validated method of increasing cholesterol biosynthetic flux by culturing cells in serum-free media[24], which results in upregulation of the entire pathway, primarily by the action of SREBP2 transcription factor. SQLE-sensitive cell lines displayed significantly accentuated sensitivity to NB-598 treatment under these conditions (Supplementary Fig. 8a). Interestingly, SQLE-insensitive cell lines displayed NB-598 sensitivity in the context of serum-free media (Fig. 6e). As expected, treatment of cells with NB-598 in the serum-free media resulted in higher levels of accumulated squalene (Fig. 6f). To exclude the possibility that experiments in serum-free media reflected a generic requirement for cholesterol needed for growth, we included an additional pathway inhibitor, atorvastatin, which showed no significant effects on cellular proliferation (Supplementary Fig. 8b). It is important to note that differential sensitivity between various cholesterol pathway inhibitors was only observed in experiments confined to ≤72 h. As expected, longer treatments of all SCLC cell lines with cholesterol pathway inhibitors in the context of serum-free media adversely affected cellular growth, likely reflecting general cholesterol auxotrophy (data not shown). To confirm that the cholesterol pathway was inhibited to similar extent regardless whether NB-598 or atorvastatin were used, we conducted $^{13}C_2$-acetate labeling experiments. While atorvastatin-treated H2081 and H82 cells lines displayed some residual cholesterol biosynthesis, the results between atorvastatin- and NB-598-treated H2171 cells were indistinguishable (Supplementary Fig. 8c).

Collectively, these results are consistent with a multifactorial model where squalene accumulation is necessary (Fig. 5) but not sufficient (Fig. 4d) for the induction of the growth defects in a subset of SCLC cells. The cellular ability to sequester squalene in lipid droplets appears to be protective (Fig. 6d) and may be one of multiple factors contributing to differential responses observed between SQLE-sensitive and SQLE-insensitive cell lines upon NB-598 treatment.

## Discussion

SCLC accounts for approximately 13–25% of all lung cancer cases and represents a particularly aggressive and hard-to-treat subset of lung cancers[25]. Current therapies heavily rely on chemotherapy, and while most SCLC patients initially respond to the platinum and etoposide regimen, relapse is exceedingly common. The paucity of therapeutic options is to some extent explained by the complex genetics of this disease and the lack of classically defined druggable oncogenic-driver genes. SCLC tumors are characterized by almost universal biallelic loss of RB transcriptional co-repressor 1 (*RB1*) and tumor protein p53 (*TP53*)[26]. Some of the less frequent genomic alterations include amplification of *SOX2*, *BCL2*, *MYCL1*, *MYCN*, and *MYC*, as well as loss-of-function mutations in *NOTCH* family genes. The functional consequences of specific genomic aberrations on disease progression and clinical outcomes are not fully understood. Alternative approaches to address SCLC are clearly needed, as evidenced by robust efforts to define new vulnerabilities using existing pharmacological agents[27,28], identify new combination therapies[29], and discover novel therapeutic targets[30].

*SQLE* locus displays increased copy number alterations in multiple cancers, based on its proximity to *C-MYC* on chromosome 8q24.1, and has been proposed as a cancer target[31]. SQLE is also overexpressed in nonalcoholic fatty liver disease-induced hepatocellular carcinoma and has been proposed as a cancer target in that context[32]. While SQLE overexpression can be an independent predictor of worse prognosis in breast cancer[31], our pattern of sensitivity to SQLE inhibition is entirely novel and is not related to any of the previously identified stratification criteria. The sensitivity appears to be enriched among the ASCL1-high/NEUROD1-low subset of SCLC cell lines, suggesting a unique biological context associated with this metabolic vulnerability. Previous efforts to further define SCLC subtypes have identified three distinct subsets based on the expression of key lineage-specifying transcription factors: ASCL1-high/NEUROD1-low, ASCL1-low/NEUROD1-high, and the smallest subset, ASCL1-low/NEUROD1-low[10]. While the clinical implications of these findings are still not clear, characterization of pre-clinical models increasingly suggests that these subsets represent distinct biological entities with discrete therapeutic opportunities, such as targeting DLL3, a direct target of ASCL1[10], in ASCL1/DLL3-high tumors[33], targeting aurora kinase A in NEUROD1-high tumors[34], or targeting IMPDH in ASCL1-low tumors[35]. Our results further strengthen these observations by demonstrating

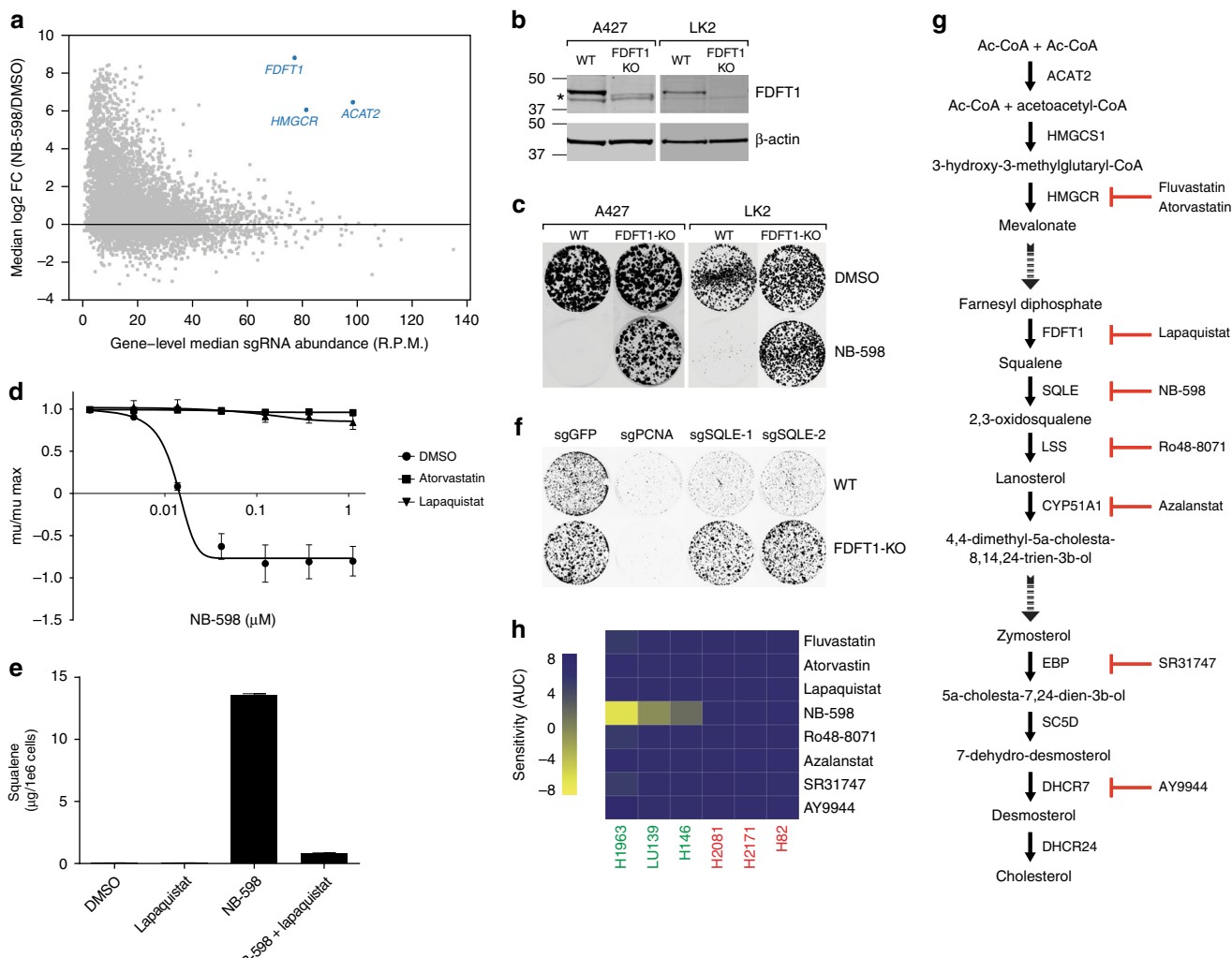

**Fig. 5** Squalene accumulation is necessary for NB-598 efficacy. **a** A CRISPR screen identifies multiple steps in the cholesterol biosynthetic pathway as suppressors of NB-598 efficacy. The median log2-fold changes of targeting sgRNAs between NB-598 and DMSO control are plotted for all genes. The gene-level sgRNA abundance is plotted in reads per million (R.P.M.). **b** Immunoblot analysis showing no full-length FDFT1 expression in the FDFT1-KO cells in comparison to wild-type cells (A427 and LK2). Asterisk (*) indicates a non-specific band. Molecular weight markers are indicated on the left in kDa units. **c** Suppression of NB-598 growth defects in FDFT1-KO clones in multiple independent cell lines (A427 and LK2). Cells were treated with 1 μM NB-598. Colony-formation assays were conducted in six-well plates. **d** Pharmacological suppression of NB-598 growth defects in LU139 cells using 10 μM atorvastatin or 10 μM lapaquistat. Mean values of triplicate measurements from a representative experiment are plotted and error bars represent s.d. **e** Treatment with 10 μM lapaquistat suppresses squalene accumulation induced by 1 μM NB-598 in LU139 cells. Mean values of triplicate measurements from a representative experiment are plotted and error bars represent s.d. **f** CRISPR-mediated KO of SQLE leads to growth defects in LK2 cells that are suppressed in the context of FDFT1-KO clone. sgRNA targeting an essential gene, *PCNA*, is used as a control. Colony-formation assays were conducted in six-well plates. **g** Schematic representation of the cholesterol biosynthetic pathway illustrating the site of action of multiple cholesterol pathway inhibitors. Simplified view of the pathway is shown, not accounting for multiple known bifurcating steps, such as di-oxidosqualene formation by SQLE or lanosterol to 24,25-dihydrolanosterol formation by DHCR24. **h** Relative sensitivities of a SCLC cell line panel to multiple inhibitors in the cholesterol pathway. AUC values derived from the mu/mu.max curves are shown. Top dose for each of the inhibitors was 3 μM

additional lineage-specific vulnerabilities that can be exploited for therapeutic interventions.

The cholesterol biosynthetic pathway has been extensively studied and statins—inhibiting the rate controlling step of the pathway, HMGCR—are currently used to treat hypercholesterolemia[36]. The widespread use of these drugs has enabled epidemiological observations of lowered cancer incidence[37], but the mechanistic details are still not fully understood. Cholesterol pathway–associated tumor vulnerabilities are being actively explored and several tumor genotypes have been particularly implicated. Mutant p53 breast cancers appear to upregulate the entire pathway and display sensitivity to its inhibition[38]. Mutant

PTEN prostate cancers appear to accumulate cholesterol esters and their depletion results in significant reduction of tumor cell growth[39]. Our identification of a subset of SCLC cell lines sensitive to SQLE inhibition is unique, as these cells do not display sensitivity to other inhibitors in the cholesterol biosynthetic pathway. Instead, the accumulation of squalene, or any potential squalene-derived metabolites, appears to be toxic to this subset of SCLC cell lines.

Squalene appears to be stored in specialized compartments called lipid droplets, which are predominantly composed of neutral lipids such as triacylglycerides and sterol esters. In addition to serving as lipid storage depots, there is growing

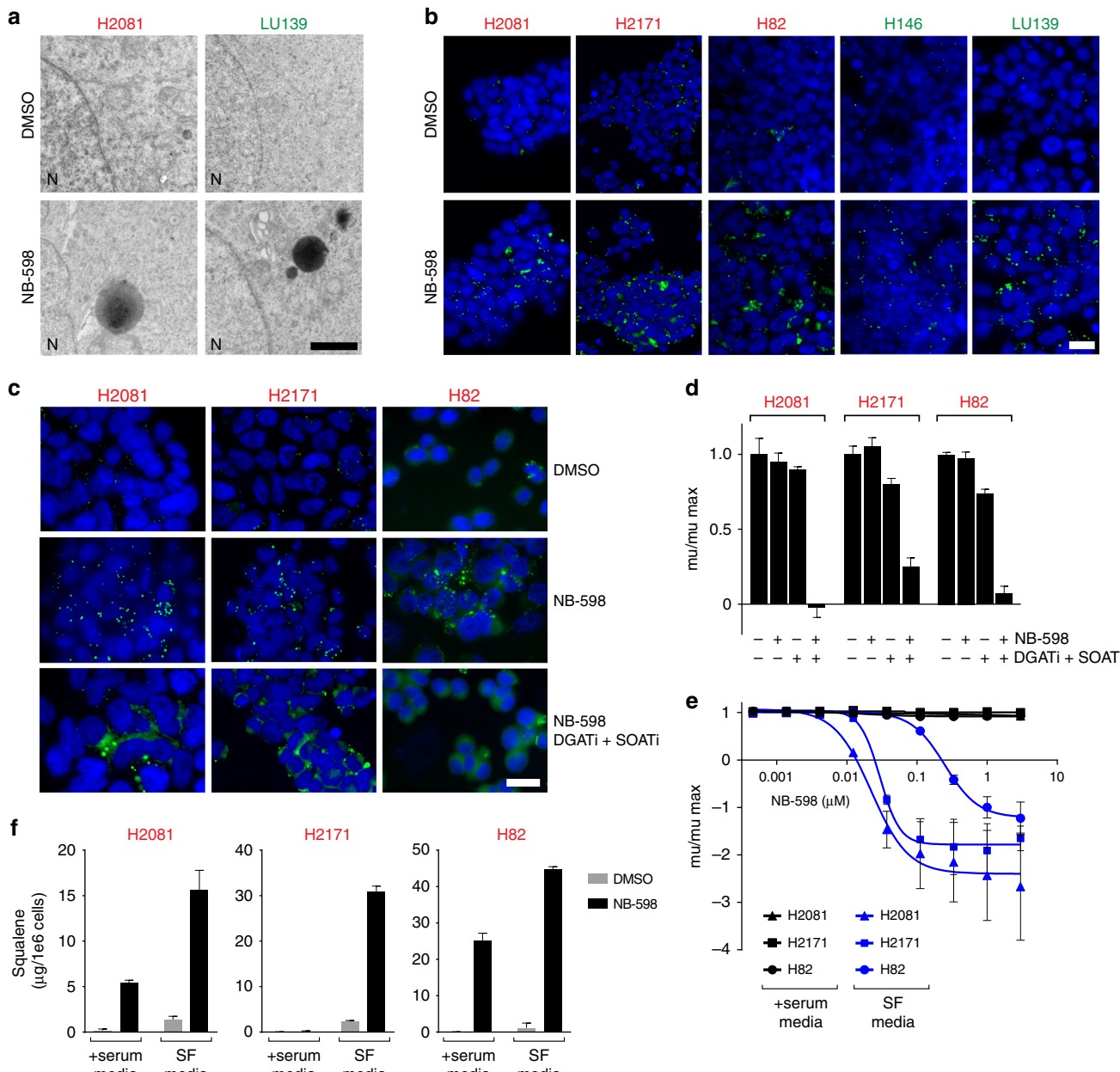

**Fig. 6** Lipid droplets and squalene storage. **a** Transmission electron microscopic image of darkly staining droplets. LU139 and H2081 cells were treated with 1 μM NB-598 for 48 h. Scale bar represents 1 μm. **b** BODIPY neutral lipid stain identifies lipid droplets in multiple cell lines. Cells were treated with 1 μM NB-598 for 24 h. Scale bar represents 20 μm. **c** Inhibition of DGAT and SOAT enzymes abrogates the formation of NB-598-induced lipid droplets. Cells were treated for with 1 μM NB-598. AZD-3988 was used as the DGAT inhibitor (10 μM for H2081 and H2171 cell lines; 7.5 μM for H82 cell line). F12511 was used as the SOAT inhibitor (10 μM for H2081 and H2171 cell lines; 7.5 μM for H82 cell line). Treatment duration was 24 h. Scale bar represents 20 μm. **d** Relative NB-598 sensitivity of SQLE-insensitive cell lines after abrogation of lipid droplet formation. Inhibitor concentrations used were identical to the ones used in the preceding panel. Treatment duration was 72 h. **e** Relative NB-598 sensitivity in a representative panel of SQLE-insensitive cell lines grown in two different media: serum-containing and serum-free (SF). **f** Squalene accumulation in cell pellets in a panel of SCLC cell lines after 1 μM NB-598 treatment for 24 h. For **d**–**f**, mean values of triplicate measurements from a representative experiment are plotted and error bars represent s.d

recognition that lipid droplets have additional functions, such as protection against lipotoxicity, regulation of autophagy, and control of energy and redox homeostasis[40]. Our studies showed that squalene was stored in lipid droplets in all SCLC cell lines characterized. However, the abrogation of lipid droplet formation resulted in dramatic induction of NB-598 sensitivity in the SQLE-insensitive cells. In fact, sensitivity of all cells to NB-598 could be further enhanced by increasing the cholesterol pathway flux, as

demonstrated by experiments utilizing serum-free media, suggesting that cellular squalene storage capacity is an important parameter to consider and that there may exist inherent differences among cell lines. While the exact mechanisms involved remain to be elucidated, a recent study identified CASIMO1, a novel small protein that interacts with SQLE and that was able to influence lipid droplet formation[41]. Future studies will focus on characterization of differential cellular responses to accumulating

squalene in the context of SQLE-sensitive and SQLE-insensitive cells. Identifying molecular mechanisms through which squalene exerts its effects on cellular growth will be of high interest, as such findings may have implications for broader cellular capability to handle additional classes of lipids and thus may lead to the identification of additional therapeutic vulnerabilities in the SQLE-sensitive subset of SCLC cell lines.

Interestingly, our findings bear some similarity to the vulnerability of specific fungi to the inhibition of fungal SQLE by the approved drug terbinafine, where sensitivity is not thought to be related to general sterol depletion, and similarly, the mechanism of cellular toxicity remains to be elucidated[42]. Recent report in *Saccharomyces cerevisiae* identified enhanced terbinafine sensitivity in yeast strains lacking the ability to assemble lipid droplets, suggesting that these structures may serve as depots for squalene under conditions of SQLE inhibition[43]. Further studies will be needed to determine whether similar mechanisms underlie the sensitivity of SCLC cells to SQLE inhibition and whether additional therapeutic targets could be identified to exploit this vulnerability.

A growing body of evidence suggests that in vivo–grown tumors are metabolically flexible in terms of the fuels or biosynthetic pathways they utilize[44] and thus posing a significant challenge to therapeutic strategies aimed at limiting a particular metabolite. Therefore, approaches focused on exploiting vulnerabilities associated with the accumulation of specific metabolites present an appealing alternative. For example, frequent deletion of the enzyme methylthioadenosine phosphorylase (MTAP), due to its proximity to p16/CDKN2A tumor-suppressor gene, results in accumulation of the metabolite MTA, which significantly reduces the activity of PRMT5 methyltransferase, thereby creating a therapeutic opportunity[45–47]. To our knowledge, the unambiguous case of a metabolic intermediate building up to toxic levels and leading to efficacy in specific tumor types has not been documented. However, it is worth noting that the concept of a toxic metabolic intermediate is actively exploited in the clinic in the context of photodynamic therapy, wherein the local application of 5-aminolevulenic acid (ALA) results in the significant buildup of protoporphyrin IX (PPIX), an effective photosensitizer[48]. Ferrochelatase (FECH) catalyzes the conversion of PPIX to heme, a pathway end product without photosensitizer properties, but the local application of ALA exceeds the capacity of FECH, thus leading to therapeutically exploitable accumulation of PPIX.

Collectively, our results highlight SQLE as a potential therapeutic target in a subset of neuroendocrine tumors, particularly SCLC. In contrast to the well-appreciated approach of blocking the overall output of various metabolic pathways as a potential therapeutic avenue, we demonstrate that additional vulnerabilities that rely on the accumulation of toxic intermediates can occur in specific cancer subsets. More broadly, our findings expand the repertoire of potential mechanisms that can be exploited in targeting metabolic nodes for precision cancer therapy.

## Methods

**Large-scale screen (482 cancer cell lines)**. Cells were seeded in recommended growth media in black 384-well tissue culture treated plates at 500 cells per well. Cells were equilibrated in assay plates via centrifugation and placed in incubators for 24 h before treatment. At the time of treatment, a set of assay plates (which did not receive treatment) was collected and ATP levels were measured by adding ATPLite (Perkin Elmer) on Envision Plate Readers. Treated assay plates were incubated with compounds for 72 h and subsequently were developed for endpoint analysis using ATPLite. Each compound was assessed using a nine-point dilution curve. Cells were defined as sensitive to a given compound based on GI75 < 2 μM with manual quality control of curve fit. GI75 is defined as a drug concentration that causes a 75% reduction in assay readout relative to dimethyl sulfoxide (DMSO)-treated controls.

**SCLC cell lines**. SCLC cell lines were obtained from the American Type Culture Collection (ATCC), with the following exceptions: LK2 cells were obtained from the Japanese Collection of Research Bioresources Cell Bank (JRCB), LU139 cells from Riken BRC, and SCLC-21H cells from Deutsche Sammlung von Mikroorganismen und Zellkulturen GmbH (DSMZ). Dulbecco's modified Eagle's medium (DMEM)/F12 media (11320033, Thermo Fisher Scientific) supplemented with 10% fetal bovine serum (F2442, Sigma) in a humidified atmosphere (5% $CO_2$) at 37 °C was used for all experiments involving SCLC cell lines. ACL4 medium was used in the experiments requiring serum-free medium.

All cell lines were routinely assessed for mycoplasma using the MycoAlert Mycoplasma Detection Kit (LT07–418, Lonza) and tested negative.

**Drug sensitivity in the 42 SCLC cell line panel**. For each cell line, cell growth was measured in triplicate across a ten-point dose curve of NB-598 or other cholesterol pathway inhibitor. The parameter mu was calculated as the log of the ratio of the CellTiter-Glo signal at 96 h to the signal at 0 h, and mu.max for each cell line was defined as the average mu value corresponding to DMSO-treated cells. For each cell line, a sigmoid function was fit to mu/mu.max as a function of log([drug]), and the AUC was defined as the integral of the fitted curve. For categorical analysis, AUC values <2.0 were considered sensitive, AUC values between 2.0 and 9.0 were considered moderate, and AUC values >9.0 were considered insensitive.

**Individual cell viability assays**. Cells were plated at $1 \times 10^4$ cells per well for suspension cell lines and $2 \times 10^3$ cells per well for adherent lines in 96-well plates (in triplicates) and were treated with increasing concentrations of inhibitors. CellTiter-Glo (CTG) cell viability assay was performed according to the manufacturer's instructions (G7570, Promega) at $T = 0$ and at the end of the experiment, $T = 96$ h. An ATP standard curve was used for every CTG readout to normalize for batch-to-batch variation and slight deviations in incubation times, thus allowing rigorous comparisons of fold growth during the course of the assay. Growth rates (mu) were calculated using the following formula:

$$mu = LN(Tend\text{-}blank)/(T0\text{-}blank)]/time(h)$$

Mu/mu.max calculations were used to compare growth rates of drug-treated to DMSO-treated cells (mu.drug/mu.DMSO), where maximum growth is observed in DMSO-treated cells. Value of mu/mu.max = 1 corresponds to no effect of drug added. Values of mu/mu.max < 0 denote cytotoxic effects (fewer cells at Tend than T0). Values of mu/mu.max between 0 and 1 denote various extents of partial growth inhibition.

**Small molecule inhibitors**. The following compounds were purchased from commercial vendors: fluvastatin (S1909, Selleck Chemicals), atorvastatin (S2077, Selleck Chemicals), lapaquistat (SC-488705, Santa Cruz Biotechnology), Ro48–8071 (10006415, Cayman Chemical), AY 9944 (14611, Cayman Chemical), NB-598 (N4536, Sigma), and AZD-3988 (4837, Tocris).

Azalanstat, SR31747, and F12511 were synthesized using previously published methods[19,49].

**Generation of FDFT1 KO clones**. Cas9 protein was transiently transfected into target cells using Lipofectamine CRISPRMAX reagent (CMAX00001, Life Technologies) followed by transient transfection of an in vitro transcribed sgRNA targeting exon 6 (ACGGCCAAGTCAATATTCTC) using RNAiMAX (13778075, Life Technologies). Forty-eight hours post transfection, single cells were selected for clonal expansion and target gene KO was assessed by FDFT1 antibody.

**sgSQLE experiments**. Cells expressing constitutive Cas9 protein were generated using the pRCCB-CMV-Cas9–2A-Blast plasmid (SVC9B-PS, Cellecta) packaged into VSV-G pseudotyped lentiviral particles. Cells were subsequently infected with sgRNA-expressing lentiviral vectors. After 48 h puromycin selection, cells were plated for colony-formation assays (see below).

Sequences of the sgRNAs used:
sgGFP: AAGATCGAGTGCCGGCATCAC
sgPCNA: CCAGGGCTCCATCCTCAAGA
sgSQLE-1: GAAAACAATCAAGTGCAGAG
sgSQLE-2: GCAGCTGTGCTTTCCAGAGA

**Immunoblotting**. Cells were washed once with 1× phosphate-buffered saline (PBS) and harvested in 1× RIPA buffer (BP-115, Boston Bioproducts) containing phosphatase and protease inhibitor cocktail (5872S, Cell Signaling Technologies). Cell lysates were briefly sonicated and subsequently cleared by centrifugation at 14K rpm for 10 min at 4 °C. Protein quantification was done by Pierce BCA Assay (23225, Life Technologies). For immunoblotting analysis, lysates were loaded onto precast sodium dodecyl sulfate-polyacrylamide gel electrophoresis gels (5671093, Bio-Rad) and subsequently transferred onto nitrocellulose membrane for detection. All primary antibodies were probed overnight at 4 °C, and membranes were washed with TBST and incubated with appropriate secondary antibodies for 1 h. Subsequently membranes were washed with TBST and visualized using Odyssey imaging system (LI-COR).

Primary antibodies used were FDFT1 (13128–1-AP, Proteintech, 1:1000), and β-actin (3700S, Cell Signaling, 1:5000). Secondary antibodies used were IRDye 680RD Donkey anti-Rabbit (926–68073, LI-COR, 1:5000) and IRDye 800CW Donkey anti-Mouse (926–32212, LI-COR, 1:5000).

Uncropped images are provided as Supplementary Fig. 9.

**Lentiviral techniques**. Lentivirus-based constructs were made using the standard protocol from The RNAi Consortium protocol from the Broad Institute (http://portals.broadinstitute.org/gpp/public/resources/protocols).

**qPCR analyses**. RNA was extracted using the Qiagen RNeasy Plus Mini Kit according to the manufacturer's protocol (74136, Qiagen). cDNA was synthesized by converting extracted RNA using the High Capacity cDNA Synthesis Kit from Applied Biosystems according to the manufacturer's protocol (4368813, Applied Biosystems). Relative gene expression levels were monitored using the following Taqman assays from Applied Biosystems:

HMGCR (Hs00168352_m1)
FDFT1 (Hs00926054_m1)
SQLE (Hs01123768_m1)
LDLR (Hs01092524_m1)
ASCL1 (Hs00269932_m1)
NEUROD1 (Hs01922995_s1)

Reactions used Advanced Fast Master Mix (4444557, Applied Biosystems) and CT values were normalized to β-actin (4326315E, Applied Biosystems), as the endogenous control.

**CRISPR screen**. A427 cells expressing constitutive Cas9 protein were generated using the pRCCH-CMV-Cas9–2A-Hygro plasmid (SVC9-PS, Cellecta) packaged into VSV-G pseudotyped lentiviral particles. A custom sgRNA library (Cellecta), containing 52,331 sgRNAs targeting 6461 genes, was similarly packaged into lentiviral particles. A427-Cas9 cells were infected with the sgRNA lentiviral library at multiplicity of infection = 0.3 and a cell density sufficient to maintain a minimum of $5.5 \times 10^7$ cells after 24 h of puromycin selection. After puromycin selection, the cells were cultured in the presence of 50 nM NB-598 or DMSO. Media was changed every 2 days until a minimum of $1 \times 10^7$ cells was reached in the NB-598-treated samples. Control samples (treated with 50 nM NB-598 but without the sgRNA library) were completely killed by drug treatment during the course of the experiment.

Upon completion of the screen, genomic DNA was extracted using phenol: chloroform method, followed by ethanol precipitation, and the sgRNA barcodes were amplified using the following primers:

FwdU6–3: ATTAGTACAAAATACGTGACGTAGAA
R2: AGTAGCGTGAAGAGCAGAGAA

The PCR product was sequenced using the Hiseq 4000 Sequencing Platform with 80–100 million single-end 50 bp reads per sample.

The constant fixed sequence immediately before 22 nucleotide barcode sequences was first removed before read mapping using AWK command in Unix. To determine the read count value for each sgRNA, the trimmed barcode reads were next mapped to custom sgRNA reference library using Bowtie2 (version 2.1.0) with option -L 22 -N 0 -k 1. These criteria retained only the perfectly matched reads for downstream analysis. The raw read count was normalized to the library size by dividing the read count by the total number of mapped reads.

For each sgRNA, log2-fold change of normalized read count between NB-598- and DMSO-treated conditions was calculated. For each gene, the median log2-fold change of its targeting sgRNAs between NB-598 and DMSO condition was determined to represent gene-level enrichment or depletion. The gene-level sgRNA abundance is calculated by taking the sum of the median sgRNA abundance of NB-598 and DMSO conditions in reads per million mapped reads. The gene-level median log2-fold change vs sgRNA abundance is plotted in a Bland–Altman MA plot. The sgRNA count was calculated using a custom script in Python (version 2.7.3) and Bland–Altman MA plot was generated using R (version 3.1.1). Additional information will be made available upon request.

**Colony-formation assays**. A427 and LK2 FDFT1-KO cells were plated at $1 \times 10^4$ cells per well in 6-well culture dishes in RPMI-1640 media (12–702F, Lonza) with 10% FBS (F2442, Sigma). After 24 h, the cells were treated with either 1% DMSO or 1 μM NB-598. After 10 days the cells were washed with PBS, stained with .5% crystal violet (C581, Fisher Scientific) in a 4% paraformaldehyde solution (F8775, Sigma) and then visualized using Odyssey imaging system (LI-COR).

**Caspase activity assay**. Cells were plated at a density of $1 \times 10^4$ cells per well in 96-well plates and treated with increasing concentrations of NB-598. DMSO 0.1% treatment was used as a control. Caspase activity was assessed at using Caspase-Glo 3/7 assay (G8090, Promega) following the manufacturer's recommendations. Briefly, the reagent was added to the cells at a 1:1 ratio. The plates were then incubated at room temperature for 1 h, after which they were analyzed for luminescence in a luminometer. Values were normalized to 0.1% DMSO control after subtracting background signal.

**BODIPY lipid droplet staining**. Cell lines were plated on poly-D-lysine coverslips (1232B51, Thermo Fisher) in 24-well plates and treated with the indicated pharmacological inhibitors for 24 h. Following the treatment, cells were spun down for 5 min at 2500 rpm. The media was aspirated and the cells were carefully washed twice with PBS. BODIPY 493/503 neutral lipid dye (D3922, Thermo Fisher) was diluted in PBS to working concentration of 2 μM. Cells were then incubated in 1 mL staining solution for 15 min at 37 °C. Following staining, cells were carefully washed twice with PBS, spun down for 5 min at 2500 rpm, and then fixed in 4% paraformaldehyde (50–980–495, Fisher Scientific) for 30 min at room temperature. Following fixation, cells were carefully washed twice with PBS, and the coverslips were transferred onto glass microscope slides treated with VECTASHIELD Antifade Mounting Medium with 4,6-diamidino-2-phenylindole (H-1200, Vector Labs). Images were analyzed using the Image J software.

**Transmission electron microscopy**. Electron microscopy was performed in the Microscopy Core of the Center for Systems Biology/Program in Membrane Biology (Massachusetts General Hospital, Boston).

Human cell lines grown in suspension were fixed in 2.0% glutaraldehyde in 0.1 M sodium cacodylate buffer, pH 7.4 (Electron Microscopy Sciences), rinsed in 0.1 M sodium cacodylate buffer, scraped, and pelleted. Pellets were post-fixed in 1.0% osmium tetroxide in cacodylate buffer for 1 h, rinsed several times in cacodylate buffer, and stabilized in 2.0% agarose in PBS. The agarose blocks containing the pellets were dehydrated through a graded series of ethanol solutions to 100%, followed by a brief dehydration step in 100% propylene oxide. Specimens were then allowed to infiltrate in a 1:1 solution of Eponate resin (Ted Pella) and propylene oxide overnight on a gentle rocker at room temperature. The following day, specimens were transferred into fresh 100% Eponate resin and allowed to infiltrate for several hours, then embedded in flat molds in 100% Eponate, and resin allowed to polymerize 24–48 h at 60 °C.

Ultrathin (70 nm) sections were obtained using a diamond knife (ultra-grade, Diatome) and a Leica EM UC7 ultramicrotome and collected onto formvar-coated grids (Electron Microscopy Sciences). Thin sections were stained with uranyl acetate and Reynold's lead citrate and examined in a JEOL JEM 1011 transmission electron microscope at 80 kV. Images were collected using an AMT digital imaging system with proprietary image capture software (Advanced Microscopy Techniques).

**Xenograft studies**. Five-to-6-week-old female mice were obtained from either Taconic Laboratories (ICR SCID), Jackson Laboratories (NSG) or Charles River Laboratories (Nu/Nu) and maintained in ventilated caging. Experiments were carried out under an Institutional Animal Care and Use Committee–approved protocol, and institutional guidelines for the proper and humane use of animals were followed. Xenografts (H1963 cells in ICR SCID mice; H446 cells in Nu/Nu mice) were established by subcutaneously injecting 100 μL of cells at a concentration of 5e6 per mouse in the right flank of each mouse. Similarly, xenografts (LU139 and H2171 cells in NSG mice; H146 cells in ICR SCID mice) were established by subcutaneously injecting 100 μL of cells at a concentration of 10e6 cell per mouse also into the right flank of each mouse. All cell lines were prepared for implantation by resuspending cells in DMEM/F12 and Matrigel at a 1:1 ratio. Upon establishment of tumors (150–250 mm³), mice were randomized into two groups: vehicle (0.5% methylcellulose, 4% HPMC-AS) or NB-598 (300 mg kg$^{-1}$). Each group received treatment once daily via oral administration at 10 mL kg$^{-1}$. Tumor volume was measured twice weekly by caliper, and volume was calculated using the formula $0.5 \times W^2 \times L$ with the results presented as means and standard error of measurement (s.e.m.). At the end of the study, mice were sacrificed by $CO_2$ inhalation; mice were bled via cardiac puncture, and blood sampled into EDTA tubes. Blood samples were spun at 10,000 rpm for 5 min and plasma decanted into labeled Eppendorf tubes. Tumors were aseptically removed and flash frozen with liquid nitrogen and both plasma and tumor were stored at −80 °C until analysis.

For in vivo $D_2O$-labeling studies, mice bearing LU139 xenografts (tumor volume between 300 and 600 mm³) were dosed with three daily treatments of vehicle or NB-598 to reach a steady-state inhibitor concentrations. At 24 h post last dose, animals received an intraperitoneal bolus injection of 19.8% $D_2O$ at 20 mL kg$^{-1}$. Tumors were harvested after 6 h of $D_2O$ administration.

**Immunohistochemistry**. Frozen tumors were sectioned and stained with the indicated antibodies using standard methods (Tufts Discovery Pathology, Cummings School of Veterinary Medicine).

**In vitro labeling studies**. For in vitro $D_2O$ labeling, cells were grown in the DMEM/F12 media and supplemented with 0.5% $D_2O$ (151882, Sigma) for the indicated amounts of time.

For in vitro $^{13}C_2$-acetate labeling, cells were grown in DMEM/F12 media and supplemented with 500 μM $^{13}C_2$-sodium acetate (CLM-440–1, Cambridge Isotope Laboratories) for 48 h.

**MS analysis of cholesterol pathway**. For quantitative squalene analysis, samples were quenched with a mixture of methanol and water (80:20, v/v) and extracted with ethyl acetate containing stable labeled internal standard. The supernatant was

dried down under nitrogen and then reconstituted with 100 μl of 0.4 mg mL$^{-1}$ BHT in acetonitrile (ACN). An aliquot of 20 μl was injected into the ultra-performance liquid chromatography tandem mass spectrometry (UPLC-MS/MS) system. The instrument setup consisted of an AB Sciex API-4000 QTrap Mass Spectrometer (AB Sciex, Framingham MA, USA) equipped with a Waters UPLC Acquity (Waters, Milford, MA, USA). The UPLC separation was performed on an ACQUITY UPLC BEH C18 (2.1 × 50 mm$^2$, 1.7 μm, Waters) at 40 ℃. Formic acid in water (0.1%, v/v, mobile phase A) and a mixture of ACN and isopropanol (80:20, v/v with 0.1% formic acid, mobile phase B) were employed as the mobile phase. A gradient elution over 4 min with flow rate set at 0.6 mL min$^{-1}$ was used for chromatographic separation. Squalene and 2,3-oxidosqualene were ionized under a positive ion spray mode via atmospheric chemical ionization (APCI) and detected through the multiple-reaction monitoring of a mass transition pair at $m/z$ 411.3 → 95.1 and 427.4 → 409.5, respectively.

For $^{13}$C isotopomer analysis of the cholesterol pathway, cells were extracted with ethyl acetate and analyzed via UPLC high-resolution MS. The instrument set up consisted of a Thermo QExactive Mass Spectrometer (Thermo, San Jose, CA, USA). The UPLC separation was performed on an ACQUITY UPLC BEH C18 (2.1 × 50 mm$^2$, 1.7 μm, Waters) at 40 ℃. Formic acid in water (0.1%, v/v, mobile phase A) and a mixture of ACN and isopropanol (80:20, v/v with 0.1% formic acid, mobile phase B) were employed as the mobile phase. An isocratic elution of 98% mobile phase B was used and run time was 1.5 min. The flow rate of mobile phase was set at 0.6 mL min$^{-1}$. Squalene and 2,3-oxidosqualene were ionized under a positive ion spray mode via APCI. Peak areas were calculated in El Maven (https://elucidatainc.github.io/ElMaven/) and stable isotopic measurements were corrected against the naturally occurring isotopes for each metabolite measured.

For $^2$H isotopomer analysis of the cholesterol pathway, cells were extracted from cells or tissues with 100 μL 1 N KOH in 80% ethanol. Samples were heated for 65 ℃ for 1 h. Thereafter, the pH of the samples were adjusted to below pH 2 with the addition of 6 N HCl (25 μL). Lipids were extracted with 150 μL chloroform and centrifuged for 10 min (2000 × g). The chloroform layer was removed and evaporated to dryness. Acetate esters of cholesterol was prepared by adding 100 μL acetic anhydride:pyridine (2:1, v/v) to the dried samples. Samples were heated for 1 h at 75 ℃. Reagents were evaporated slowly by nitrogen gas. Samples were reconstituted with 100 μL with ethyl acetate and transferred to autosampler vial for analysis.

Cholesterol ester preparations were analyzed for deuterated-palmitate with a Thermo Finnigan Delta V IRMS coupled to a Thermo Trace GC Ultra with a GC combustion interface III and Conflow IV. The acetate ester of $^2$H-cholesterol was analyzed using a split-less injection with CTC Pal autosampler (1 μL), at an injection temperature of 250 °C, and using a Zebron ZB-5 column of 30 m × 0.25 mm × 0.50 μm film thickness (Phenomenex, Torrance, CA). The GC oven was programmed with an initial column temperature of 100 °C with a 1-min hold, followed by a ramp of 10 °C per minute to 150 °C and a final ramp of 30 °C min$^{-1}$ to 340 °C. Compounds eluting off the column were directed into the pyrolysis reactor, heated at 1450 °C, and converted to hydrogen gas. The deuterated enrichment was first initially expressed in delta values compared to a calibrated hydrogen gas and then converted to atom % D by standard equations.

All IRMS analyses involving deuterium labeling were performed at Metabolic Solutions (Nashua, NH).

**SCLC proteomics**. SCLC cells were grown in DMEM/F12 media (11320033, Thermo Fisher Scientific) and harvested in triplicate at 5e6 cells per pellet.

Unless otherwise stated, all chemicals were from Sigma Aldrich. All water and solvents were Optima LC/MS grade from Thermo Scientific.

**Proteomics: protein extraction and digestion**. Cell pellets were lysed in 8 M urea/50 mM HEPES pH8.5 (Alfa Aesar) and treated with nuclease (Thermo Scientific) for 10 min at room temperature with constant shaking in a Thermomixer (Eppendorf). Lysates were reduced with 5 mM dithiothreitol (DTT) for 30 min at 37 ℃ and cysteine residues alkylated with 15 mM iodoacetamide for 30 min at room temperature in the dark. Excess iodoacetamide was quenched with 10 mM DTT for 15 min at room temperature in the dark. Protein was extracted by methanol−chloroform precipitation and 2× 1 mL methanol washes. Pellets were dried and resuspended in 8 M urea/50 mM HEPES, pH 8.5. Protein concentrations were measured by BCA assay (Thermo Scientific) prior to protease digestion. Two hundred micrograms of protein were diluted to 2 M urea and digested with LysC (Wako) in a 1:100 enzyme:protein ratio overnight at 30 ℃. The next morning, trypsin (Promega) was added to a final 1:100 enzyme:protein ratio for 6 h at 37 ℃. Digests were acidified with 10% trifluoroacetic acid (TFA) to a pH ~2 and subjected to solid-phase extraction (SPE) with HyperSep Retain PEP Cartridges (Thermo Scientific). Peptides were resuspended in 200 mM HEPES pH8.5/10% ACN, and concentrations were measured by microBCA assay (Thermo Scientific).

**Proteomics: tandem mass tag labeling**. Isobaric labeling of peptides was performed using 10-plex tandem mass tag (TMT) reagents (Thermo Scientific). TMT reagents (0.8 mg) were dissolved in 41 μL of anhydrous ACN and 10 μL was added to 50 μg of peptide. Samples were labeled for 2 h at room temperature and quenched by the addition of hydroxylamine to 0.5%, v/v, for 15 min. Samples were

pooled, acidified with 10% TFA to a pH ~2 and subjected to SPE. In all, 1.08 mg of peptides from 36 protein digests labeled with TMT-131 was spiked into each 10-plex to normalize across 10-plex experiments.

**Proteomics: basic pH reverse-phase high-performance liquid chromatography (HPLC) fractionation**. TMT-labeled peptides were subjected to orthogonal basic-pH reverse-phase fractionation. Peptides were solubilized in buffer A (10 mM ammonium bicarbonate, pH 8.0/5% ACN) and separated on a Biobasic C18 column (5 μm particle size, 4.6 mm ID, and 250 mm length, Thermo Scientific) using an Ultimate 3000 HPLC (Thermo Scientific) and a 44 min linear gradient from 12% to 36% ACN in 10 mM ammonium bicarbonate pH 8.0 (flow rate of 0.8 mL min$^{-1}$) into a total of 96 fractions. The 96 fractions were consolidated into 24 samples in a checkerboard manner, acidified with TFA to pH ~2, and vacuum-dried. Each sample was dissolved in 20 μL 5% formic acid (FA)/5% ACN and 2 μL was analyzed by MS.

**Proteomics: Orbitrap fusion parameters**. Spectra were acquired on an Oribtrap Fusion (Thermo Scientific) coupled to an Easy-nLC 1200 ultrahigh pressure liquid chromatography pump (Thermo Scientific). Peptides were separated on a 50 μM C18 EASYspray column (Thermo Scientific) using a 70 min, 8−28% ACN (constant 0.1% FA) gradient with a 300 nL min$^{-1}$ flow rate. MS1 spectra were collected in the Orbitrap at a resolution of 60,000, automated gain control (AGC) target of 5e5, and a max injection time of 100 ms. The 10 most intense ions were selected for MS/MS in a data-dependent manner. Precursors were filtered according to charge state (2–6 z) and monoisotopic peak assignment, and previously selected peaks were excluded using a dynamic window of 60 s with a mass error ±10 ppm. MS2 precursors were isolated with a quadrupole mass filter set to a width of 0.5 $m/z$ and detected in the ion trap operated at rapid scan rate. MS2 spectra were collected at an AGC of 1e4, max injection time of 150 ms, and CID collision energy of 35%. Synchronous precursor selection was enabled to include the top 10 MS2 fragment ions for MS3 analysis in the Orbitrap at a resolution of 60,000, AGC target of 5e5, and a max injection time of 250 ms. 50% HCD collision energy was used to ensure TMT reporter detection.

**Proteomics: data processing**. All .RAW files were processed using Proteome Discoverer 2.1.0.81. MS2 spectral assignment was performed using the SEQUEST algorithm using Uniprot Human reference proteome (UP000005640, downloaded 10/05/2016) and a list of known contaminants (CRAPome.org). Mass tolerances were set to 10 ppm for precursor ions, 0.6 Da for MS2 ions, and 20 ppm for MS3 reporter ions. MS2 false discovery rate of <1% was calculated using the Percolator algorithm. Reporter ion intensities were adjusted to correct for the isotopic impurities of the different TMT reagents (manufacturer's specifications). TMT tags on peptide N-termini per lysine residues (+229.162932 Da) and carbamido-methylation of cysteine residues (+57.02146 Da) were set as static modifications and methionine oxidation (+15.99492 Da) was set as a variable modification. Signal-to-noise values for all peptides were summed within each TMT channel and each channel was scaled according to the highest channel sum so that the sum abundance of each channel is equal. Peptides were filtered for a minimum sum signal-to-noise value of 160 across all 10 channels. Quantitation data from razor peptides were excluded and only unique peptides were used for protein quantitation.

**Statistical analysis**. Analyses were performed using the GraphPad Prism 7 software package. All data are expressed as mean of multiple measurements (n, indicating the number of replicates) from a representative experiment. Each experiment was replicated multiple times. Error bars represent s.e.m. for all in vivo studies and standard deviation (s.d.) for all the in vitro studies. Student's $t$ test was used to assess statistical significance. Exact values and cutoff are specified within each figure or figure legend.

**Code availability**. Additional statistical analyses and visualizations of cellular growth, RNA-seq, and proteomics data were performed using code written in Python using the standard NumPy, pandas, matplotlib, and seaborn packages. Code is available upon request.

**Reporting summary**. Further information on experimental design is available in the Nature Research Reporting Summary linked to this Article.

## Data availability

The mass spectrometric proteomics data have been deposited in the PRIDE data repository under the accession code PXD011896. All other data supporting the findings in this study are available within this article and supplementary files or from the corresponding author upon reasonable request.

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

## Acknowledgements

We would like to thank multiple members of the Agios scientific community for their technical assistance and helpful discussions, as well as Matthew Vander Heiden and Monty Krieger for critical review of the manuscript. This work was supported by Agios Pharmaceuticals.

## Author contributions

C.E.M., D.P., V.C., T.S., S.H., Z.P.F., E.L.A., Y.C., L.H., M.L., G.M., R.N., S.C., Y.C., S.G., G.C., A.K.P. and S.M. designed and performed the experiments and interpreted the data. Y.Z., W.L., K.M.M., J.M., M.D., S.J., N.N., S.A.B. and T.R. provided experimental expertise, data interpretation assistance, and oversight. J.P.-M. and G.A.S. co-led the project. G.A.S. wrote the manuscript with input from D.P., V.C., T.R. and K.M.M.

## Additional information

**Competing interests:** C.E.M., D.P., V.C., T.S., S.H., Z.P.F., E.L.A., G.M., R.N., S.C., Y.C., S.G., G.C., A.K.P., S.M. W.L., K.M.M., J.M., M.D., S.J., N.N., S.A.B., T.R., J.P.-M. and G.A.S. are employees of and have ownership interest in Agios Pharmaceuticals. The remaining authors declare no competing interests.

