## [Peer Review File · Nature Communications]

Reviewers' Comments:

Reviewer #1:

Remarks to the Author:

Mahoney et al report findings that reveal the metabolic enzyme squalene epoxidase (SQLE) as a significant vulnerability in a subset of tumor cells of neuroendocrine origin—most notably, neuroblastomas and small cell lung cancers (SCLC). By screening a panel of several hundred human cancer cell lines for sensitivity to various inhibitors of metabolic processes, they observed that the previously reported SQLE inhibitor NB-598 showed selective growth inhibitory activity in the neuroendocrine context. This was confirmed with in vivo studies of a few sensitive and resistant lines in xenograft studies. The identified mechanism of action, as revealed through a clever CRISPR modifier functional genomic screening approach, appears to involve the accumulation of squalene, a precursor in the biosynthesis of cholesterol. The authors conclude that these findings inform a novel therapeutic to manage SCLCs.

Overall, these are potentially important new findings, and the studies have been generally well designed. However, as detailed below, some aspects of the manuscript are confusing as currently presented, and the authors need to address a few things through additional experimentation before it is possible to adequately evaluate the significance of these observations.

Specific comments:

1. The authors seem to jump around between the use of certain cell lines for certain types of analysis in a curious and unexplained way that leaves the reader wondering whether all of the findings are relevant across all of the drug-sensitive models. While the rationale for using the adherent A427 line for the CRISPR screen is reasonable, the authors do not show IC₅₀ curves for either A427 or LK2 cells (which are suddenly introduced without explanation). Furthermore, only LU139 cells are used in the pharmacologic “rescue” study. To make the overall study more convincing, the authors need to show that in all of the sensitive cell lines used here (there aren't that many), the growth effects of NB-598 can be rescued using lapaquistat and atorvastatin. This is an easy analysis.
2. The mechanistic conclusion regarding the importance of squalene accumulation is confusing. The authors write, “No major difference were observed even in the cell lines displaying high squalene accumulation, suggesting that squalene secretion is unlikely to account for the dramatic differences in NB-598 responses.” But, the next section of the Results is titled, “Squalene accumulation is critical for NB-598 efficacy”. Overall, this seems to be a conflicting message, unless the authors are making a clear distinction between squalene secretion and intracellular accumulation. If so, this needs further attention to be convincing.
3. The authors conclude that the effects of NB-598 on SQLE are responsible for the observed efficacy in sensitive cells; however, while their findings are consistent with a requirement for SQLE inhibition in this activity, they do not demonstrate definitively that SQLE inhibition is sufficient. To reach that conclusion, the authors need to show that disruption of SQLE gene function, for example by siRNA, is sufficient to inhibit growth in the sensitive models.
4. On page 7, the authors make a curious leap between the end of one paragraph in which they describe a once-daily dose of drug to be “sufficiently to effectively inhibit SQLE in human xenograft models”, they start the next paragraph by describing a study in which they treated mice with three daily doses. This is an awkward transition.
5. On page 8, there is an oddly worded sentence: “We next hypothesized that the differential response to NB-598 might underlie the dramatically different cellular responses observed in the panel of SCLC cell lines”. Not clear exactly what is meant here.

6. In figure 2D, the y-axis is not clearly defined.

7. In figure 5C, the WT, DMSO-treated images are clearly duplicated for the two different cell lines shown. This is presumably an error in assembling the figure panels, but obviously needs to be corrected.

Reviewer #2:

Remarks to the Author:

Summary: In the manuscript by Mahoney et al. the authors perform a chemical screen using a panel of various cancer cell lines to identify sensitivity to compounds targeting various metabolic enzymes. In this screen the authors identify tumor cells originating from neuroendocrine cells (neuroblastoma and SCLC) are particularly sensitive to NB-598, which inhibits squalene epoxidase (SQLE), an enzyme in cholesterol biosynthesis. The authors go on to confirm in vitro and in vivo sensitivity to NB-598 in a subset of SCLC cell lines and investigate what determines sensitivity to NB-598. Specifically, in cells that are sensitive to NB-598, it is the accumulation of the metabolite squalene that results in cell death (apoptosis) as inhibition of enzymes in the cholesterol biosynthesis pathway up-stream of SQLE by either genetic or pharmacological means confers resistance to NB-598 and inhibition of enzymes downstream of SQLE does not reduce cell proliferation.

Major Points:

1. The authors do several experiments to characterize NB-598 sensitive cells to determine predictors of sensitivity (gene expression, proteomics, evaluation of cholesterol biosynthesis, amount of squalene accumulation). Of these experiments it seems that high levels of ASCL1 and low levels of NEUROD1 are the best predictors of NB-598 sensitivity, where all responders have this gene expression pattern. However, numerous NB-598 insensitive cell lines also have this gene expression profile. Since one of the goals of this screen is to identify synthetic lethalties to metabolic inhibitors in specific genotypes or defined groups of cancers, can the authors comment on how they would identify NB-598 responders? Would integrating expression data from the other neuroendocrine markers in Supplementary Figure 1a (CHGA, INSM1, NCAM1 etc.) with ASCL1 and NEUROD1 expression more clearly separate NB-598 sensitive from insensitive cell lines?

2. The authors provide good evidence that the accumulation of squalene is important for the efficacy of NB-598, as the inhibition of enzymes upstream of SQLE in cholesterol biosynthesis result in resistance to NB-598 and inhibition of enzymes downstream of SQLE do not show sensitivity in the cell lines tested (Figure 5g). The role of squalene as a toxic metabolite could be significantly strengthened by showing that treatment cells with squalene has the same effect as NB-598 on sensitive cell lines. Conversely, insensitive cells could potentially be sensitized to NB-598 by co-treatment with squalene. It is possible that accumulation of squalene is not toxic and just a by-product. An alternative explanation could be that SQLE has another non-metabolic function that is critical and absent upon NB-598 treatment.

3. Although the authors provide evidence that the accumulation of squalene is important for efficacy of NB-598, the amount of squalene accumulation is not correlated with sensitivity (Figure 4d). Can the authors please comment on this? Is there flexibility between cell lines in the amount of squalene that can accumulate before becoming toxic? Trying to increase squalene levels by supplementation as suggested in point 2 would clarify this issue. In insensitive cells, is there a level of squalene accumulation that results in sensitivity to NB-598?

Minor Points:

1. Please increase the color contrast in graphs with multiple groups being plotted to make data more easily interpretable. Especially in Figures 1c, 2a, and 4b-e.

Reviewer #3:

Remarks to the Author:

In this study, Mahoney and colleagues performed a drug screening to identify novel small molecules to target neuroendocrine tumors. The authors found that specific inhibitors for squalene epoxidase, the enzyme that produce the first sterol intermediate of cholesterol biosynthesis pathway suppress cellular proliferation and attenuates tumor growth in mice. Surprisingly, the authors demonstrate that the effects are mediated by the accumulation of squalene rather than the depletion of cholesterol. This finding is supported using other inhibitors of the cholesterol biosynthesis pathway including statins. While the study is of interest, the mechanism by which squalene influences cell survival and proliferation is not explored.

Major comments.

1) How and why accumulation of Squalene influences tumor cell survival and proliferation needs to be addressed. Is the plasma membrane composition affected?; How about oncogenic signaling, numerous oncogenes (RAS, etc) are located in the plasma membrane.

2) Tumor morphology should be characterized morphologically. Is the SQLE inhibitor affecting tumor cell apoptosis, proliferation, angiogenesis, tumor cell macrophages, etc. These studies are superficial.

3) The efficacy of SQLE inhibitors in regulating tumor growth in vivo should be compared to other cholesterol biosynthesis inhibitors (statins).

Overall, the authors report an interesting finding but they should provide better characterization of the in vivo tumor model and identified the molecular mechanism by which SQLE mediates cytotoxic effects in neuroendocrine tumor cell lines.

NCOMMS-18-03313A

Reviewer's comments + specific author responses:

Reviewer #1: Expert in cancer therapeutics

Mahoney et al report findings that reveal the metabolic enzyme squalene epoxidase (SQLE) as a significant vulnerability in a subset of tumor cells of neuroendocrine origin—most notably, neuroblastomas and small cell lung cancers (SCLC). By screening a panel of several hundred human cancer cell lines for sensitivity to various inhibitors of metabolic processes, they observed that the previously reported SQLE inhibitor NB-598 showed selective growth inhibitory activity in the neuroendocrine context. This was confirmed with in vivo studies of a few sensitive and resistant lines in xenograft studies. The identified mechanism of action, as revealed through a clever CRISPR modifier functional genomic screening approach, appears to involve the accumulation of squalene, a precursor in the biosynthesis of cholesterol. The authors conclude that these findings inform a novel therapeutic to manage SCLCs.

Overall, these are potentially important new findings, and the studies have been generally well designed. However, as detailed below, some aspects of the manuscript are confusing as currently presented, and the authors need to address a few things through additional experimentation before it is possible to adequately evaluate the significance of these observations.

We thank the Reviewer for their time and recognition of the novelty of our work. We address each of the individual comments below.

1.1. The authors seem to jump around between the use of certain cell lines for certain types of analysis in a curious and unexplained way that leaves the reader wondering whether all of the findings are relevant across all of the drug-sensitive models.

We apologize for the confusion. With the exception of the CRISPR suppressor screen shown in **Figure 5**, the remainder of the work shown in the manuscript is done in the same two sets of cell lines:

SQLE-sensitive (LU139, H146, H1963)

SQLE-insensitive (H82, H2171, H2081, H446)

The detailed rationale for selecting these cell lines is presented at the end of the first paragraph of the Results section and centers on our aim to adequately represent the diversity of cells that are sensitive or insensitive to SQLE inhibition.

The rationale for including two different cell lines (A427 and LK2) for the CRISPR suppressor screen centers on the fact that these cell lines are adherent and easily infected by lentiviruses. We now display NB-598 sensitivity of these cell lines to support their use as models in **Supplementary Figure 7a,b**. It is important to note that all the key results obtained from these two cell lines (suppression of NB-598 mediated growth defects by inhibition of upstream cholesterol biosynthetic steps) have been

reproduced in the canonical SQLE-sensitive cell lines used in the remainder of the manuscript (**Figure 5e** and **Supplementary Figure 7c,d**).

1.2. While the rationale for using the adherent A427 line for the CRISPR screen is reasonable, the authors do not show IC50 curves for either A427 or LK2 cells (which are suddenly introduced without explanation).

We thank the reviewer for pointing this out. We now show the associated IC50 curves for these cell lines in **Supplementary Figure 7a and 7b**. We also provide a rationale for including the LK2 cell line, which serves to strengthen our observations in another adherent SQLE-sensitive cell line.

1.3. Furthermore, only LU139 cells are used in the pharmacologic “rescue” study. To make the overall study more convincing, the authors need to show that in all of the sensitive cell lines used here (there aren’t that many), the growth effects of NB-598 can be rescued using lapaquistat and atorvastatin. This is an easy analysis.

We completely agree with the Reviewer about the importance of these studies. In fact, these exact experiments were included in **Supplementary Figure 2** before (lapaquistat and atorvastatin rescue experiment in two additional SQLE-sensitive cell lines – H146 and H1963). We now emphasize this point more in the text of the revised manuscript. In the updated manuscript, these results are now presented as **Supplementary Figure 7c and 7d**.

2. The mechanistic conclusion regarding the importance of squalene accumulation is confusing. The authors write, “No major difference were observed even in the cell lines displaying high squalene accumulation, suggesting that squalene secretion is unlikely to account for the dramatic differences in NB-598 responses.” But, the next section of the Results is titled, “Squalene accumulation is critical for NB-598 efficacy”. Overall, this seems to be a conflicting message, unless the authors are making a clear distinction between squalene secretion and intracellular accumulation. If so, this needs further attention to be convincing.

We thank the Reviewer for pointing out the potentially confusing wording. Our conclusion is that squalene accumulation is necessary, but not sufficient, to result in growth defects in multiple SQLE-sensitive cell lines. We now make this point more clearly throughout the revised manuscript. We have also included additional dataset focusing on the role of lipid droplets as squalene storage depot (**Figure 6** and **Supplementary Figure 8**). Additional explanations regarding this dataset are provided as a response to Reviewer #2, points 3.1-3.3.

Collectively, we hope that this sufficiently clarifies our conclusions and provides additional evidence to support them.

3. The authors conclude that the effects of NB-598 on SQLE are responsible for the observed efficacy in sensitive cells; however, while their findings are consistent with a requirement for SQLE inhibition in this activity, they do not demonstrate definitively that SQLE inhibition is sufficient. To reach that conclusion,

the authors need to show that disruption of SQLE gene function, for example by siRNA, is sufficient to inhibit growth in the sensitive models

The Reviewer points to a need to show SQLE-dependence using an alternative mode of abrogating enzyme function. To address this, we conducted a series of CRISPR-based KO experiments in LK2 cell line (**Figure 5d**). We show that multiple sgRNAs targeting SQLE result in a growth defect that can be rescued by a concomitant KO of FDFT1 (squalene synthase). This result supports the specificity of the growth defects due to SQLE loss of function in the context of SQLE-sensitive cells and is fully consistent with the other results shown in **Figure 5**.

4. On page 7, the authors make a curious leap between the end of one paragraph in which they describe a once-daily dose of drug to be “sufficiently to effectively inhibit SQLE in human xenograft models”, they start the next paragraph by describing a study in which they treated mice with three daily doses. This is an awkward transition.

We thank the Reviewer for pointing out this unclear phrasing. We took a conservative approach in our study design to ensure that the compound amount was fully equilibrated in the mouse tumors, even though this study could have been conducted with just a single dose. We now explain this more fully in the text of the revised manuscript.

5. On page 8, there is an oddly worded sentence: “We next hypothesized that the differential response to NB-598 might underlie the dramatically different cellular responses observed in the panel of SCLC cell lines”. Not clear exactly what is meant here.

We thank the Reviewer for pointing out this unclear phrasing. We now clarify this statement more in the updated manuscript. Furthermore, we provide additional data (specifically requested by Reviewer #2, points 3.1–3.3) to elaborate on the connection of squalene accumulation and different cellular responses displayed by cells. This data focusing on the relative role of lipid droplets (shown in **Figure 6**) and the associated discussion provide further context and a more detailed explanation of our original statement.

6. In figure 2D, the y-axis is not clearly defined.

The “ μ/μ_{max} ” parameter is used throughout the manuscript (**Figures 1c, 2d, 5e, 6d, 6e, S7a-d, S8a-b**) and the full explanation is provided in the methods section.

“ μ/μ_{max} calculations were used to compare growth rates of drug-treated to DMSO-treated cells (μ_{drug}/μ_{DMSO}), where maximum growth is observed in DMSO-treated cells. Value of $\mu/\mu_{max} = 1$, corresponds to no effect of drug added. Values of $\mu/\mu_{max} < 1$, denote cytotoxic effects (fewer cells at T_{end} than T_0). Values of μ/μ_{max} between 0 and 1, denote various extents of partial growth inhibition.”

As suggested by the Reviewer, we now also provide this explanation in the first figure legend this parameter is encountered.

7. In figure 5C, the WT, DMSO-treated images are clearly duplicated for the two different cell lines shown. This is presumably an error in assembling the figure panels, but obviously needs to be corrected. We sincerely apologize for this error. The correct image is now used in the revised figure.

Reviewer #2: Expert in cancer metabolism

Summary: In the manuscript by Mahoney et al. the authors perform a chemical screen using a panel of various cancer cell lines to identify sensitivity to compounds targeting various metabolic enzymes. In this screen the authors identify tumor cells originating from neuroendocrine cells (neuroblastoma and SCLC) are particularly sensitive to NB-598, which inhibits squalene epoxidase (SQLE), an enzyme in cholesterol biosynthesis. The authors go on to confirm in vitro and in vivo sensitivity to NB-598 in a subset of SCLC cell lines and investigate what determines sensitivity to NB-598. Specifically, in cells that are sensitive to NB-598, it is the accumulation of the metabolite squalene that results in cell death (apoptosis) as inhibition of enzymes in the cholesterol biosynthesis pathway up-stream of SQLE by either genetic or pharmacological means confers resistance to NB-598 and inhibition of enzymes downstream of SQLE does not reduce cell proliferation.

Major Points:

1.1. The authors do several experiments to characterize NB-598 sensitive cells to determine predictors of sensitivity (gene expression, proteomics, evaluation of cholesterol biosynthesis, amount of squalene accumulation). Of these experiments it seems that high levels of ASCL1 and low levels of NEUROD1 are the best predictors of NB-598 sensitivity, where all responders have this gene expression pattern. However, numerous NB-598 insensitive cell lines also have this gene expression profile. Since one of the goals of this screen is to identify synthetic lethalties to metabolic inhibitors in specific genotypes or defined groups of cancers, can the authors comment on how they would identify NB-598 responders?

In the initial version of the manuscript we focused on these two transcription factors, ASCL1 and NEUROD1, as they are well appreciated in the literature to define distinct SCLC subsets. These subsets, ASCL1-high/NEUROD1-low and ASCL1-low/NEUROD1-high, while defined by the differential expression levels of these two transcription factors, have been shown to display broader expression profiles that are quite distinct.

As suggested by the reviewer, we conducted broader analyses to define biomarkers associated with NB-598 response. Specifically, we performed unbiased bioinformatics analyses on RNA-Seq and proteomics datasets, in two different ways, to define expression signatures associated with SQLE sensitivity. First, we used all 42 SCLC cell lines to capitalize on the diversity of cell lines analyzed. Second, we only used 25 cell lines defined as ASCL1-high/NEUROD1-low, since all of the NB-598 responders are found within this category. We hypothesized that a more focused analysis could identify further markers associated with NB-598 response. In fact, we were able to identify such statistically significant RNA- and protein-based expression signatures in both types of analyses and we now show them as **Supplementary Figures 1-4**. Interestingly, many of the well-established neuroendocrine markers, such as CHGA and TTF1, were not present in these signatures due to the stringent cut-off criteria used. However, given the general interest in the SCLC field in these markers, we specifically display them in **Supplementary Figure 5a**.

Based on our analyses, we propose that NB-598 responders can be identified based on these expression signatures. While the results appear to be robust, we conservatively point out in the updated manuscript that further studies using many additional cell lines will be needed to confirm the ability of these expression signatures to prospectively identify NB-598 responders.

1.2. Would integrating expression data from the other neuroendocrine markers in Supplementary Figure 1a (CHGA, INSM1, NCAM1 etc.) with ASCL1 and NEUROD1 expression more clearly separate NB-598 sensitive from insensitive cell lines?

The Reviewer correctly points out that combining multiple markers is likely to improve the ability to identify NB-598 responders. To follow up on this suggestion, we have taken an unbiased approach and identified RNA- and protein-based expression signatures within 25 ASCL1-high/NEUROD1-low cell lines that are significantly associated with NB-598 response. These results are now shown in **Supplementary Figures 3 and 4**.

2. The authors provide good evidence that the accumulation of squalene is important for the efficacy of NB-598, as the inhibition of enzymes upstream of SQLE in cholesterol biosynthesis result in resistance to NB-598 and inhibition of enzymes downstream of SQLE do not show sensitivity in the cell lines tested (Figure 5g). The role of squalene as a toxic metabolite could be significantly strengthened by showing that treatment cells with squalene has the same effect as NB-598 on sensitive cell lines. Conversely, insensitive cells could potentially be sensitized to NB-598 by co-treatment with squalene. It is possible that accumulation of squalene is not toxic and just a by-product. An alternative explanation could be that SQLE has another non-metabolic function that is critical and absent upon NB-598 treatment.

The Reviewer raises a good point. We attempted to address it in several different ways:

- We directly treated cells with increasing concentrations of squalene. Unfortunately, we were not able to replicate the differential sensitivity between SQLE-sensitive and SQLE-insensitive cell lines. However, there are several important caveats to consider:
 - o Hydrophobicity of squalene. Squalene is extremely hydrophobic and its solubility is limited in aqueous solutions. Increasing amounts of added squalene resulted in formation of hydrophobic, oil-like, droplets that floated in the media and adversely affected the growth of all cell lines tested. Our attempts to somehow “formulate” squalene for delivery in aqueous solutions were not successful.
 - o Route of delivery. Our expanded work focusing on lipid droplets (displayed in **Figure 6**) showed the importance of this intracellular storage compartment. Treatment with extracellular squalene may not necessarily mimic the conditions experienced by cells upon SQLE inhibition, where squalene is accumulating intracellularly in the context of lipid droplets.
- We also conducted a series of $^{13}\text{C}_2$ -acetate tracing experiments in SQLE-sensitive and SQLE-insensitive cell lines and attempted to trace the fate of C13-carbons. While we detected robust levels of C13-labeled squalene, we did not detect any labeled metabolites unique to sensitive cells. Very low levels of C13-labeled squalene epoxides were detected in all samples. However, it

was not clear whether this was due to only partial inhibition of SQLE or due to partial oxidation of squalene during sample preparation. It's important to keep in mind that inhibition of the immediate next step in the cholesterol biosynthetic after SQLE, lanosterol synthase, results in robust accumulation of squalene epoxide, but without any adverse effects on cellular growth. Collectively, these results did not support the idea of potential toxic squalene-related metabolites. However, we cannot formally exclude such possibility and we conservatively state so in the manuscript.

3.1. Although the authors provide evidence that the accumulation of squalene is important for efficacy of NB-598, the amount of squalene accumulation is not correlated with sensitivity (Figure 4d). Can the authors please comment on this?

The Reviewer correctly points out that while squalene accumulation is necessary for NB-598 mediated growth defects, it is not sufficient. We make this point more clear now in the updated manuscript, both in the Results and the Discussion sections. Furthermore, we have expanded on the initial squalene accumulation observations by providing a series of experiments focusing on the intracellular fate of squalene and the role of lipid droplets. We were able to demonstrate that:

- Squalene appears to be stored in intracellular lipid droplets. We show this using 2 independent means – transmission electron microscopy and BODIPY neutral lipid dye. This data is shown in **Figure 6a and 6b**.
- There does not appear to be an obvious difference in the lipid droplets observed in SQLE-sensitive vs SQLE-insensitive cell lines (**Figure 6b**).
- The lipid droplet formation can be abrogated by inhibiting the synthesis of two critical lipid droplet components, triacylglycerols and sterol esters, using pharmacological inhibitors of diacylglycerol O-acyltransferases (DGATs) and sterol O-acyltransferases (SOATs), respectively. Successful abrogation of lipid droplet formation is shown in **Figure 6c**. Interestingly, in the absence of lipid droplet forming capability, representative SQLE-insensitive cell lines became susceptible to NB-598 treatment (**Figure 6d**).

3.2. Is there flexibility between cell lines in the amount of squalene that can accumulate before becoming toxic?

This is an interesting suggestion and we pursued it with some of the experiments displayed in **Figure 6**. We showed in **Figure 6a-6d** that NB-598 treated cells accumulate squalene in lipid droplets and that inhibition of lipid droplet formation can sensitize cells to NB-598 treatment. Next, we wanted to probe whether we can potentially exceed squalene storage capacity by using a well-established method of culturing cells in serum-free media, which reliably upregulates the cholesterol biosynthesis.

In fact, we were able to show that NB-598 treatment of cells in that context results in additional squalene accumulation (**Figure 6f**), which consequently results in enhanced sensitivity of SQLE-sensitive cell lines (**Supplementary Figure 8a**) and, more importantly, induction of NB-598 sensitivity in SQLE-insensitive cell lines (**Figure 6e**).

It's important to note that culturing cells in the context of serum-free media, which is devoid of lipids, and treating with inhibitors of cholesterol biosynthesis can result in generic cholesterol auxotrophy. While we do observe this phenomenon in the context of prolonged treatment times, we wanted to ensure that this is not the case in the short timeframe used for our experiments. Therefore, we compared two alternative ways of inhibiting the cholesterol pathway (atorvastatin and NB-598) in the context of serum-free media and showed that only NB-598 results in dramatic growth defects during the 72h growth experiment (**Supplementary Figure 8b**). Importantly, both inhibitors exerted roughly similar effects on the overall *de novo* cholesterol biosynthesis (**Supplementary Figure 8c**).

3.3. Trying to increase squalene levels by supplementation as suggested in point 2 would clarify this issue. In insensitive cells, is there a level of squalene accumulation that results in sensitivity to NB-598?

We thank the Reviewer for the suggestion. In the series of experiments shown in **Figure 6** and **Supplementary Figure 8**, we were able to show that NB-598 sensitivity can be induced in the context of SQLE-insensitive cells provided cholesterol pathway flux, and consequently squalene accumulation, are sufficiently increased. Details of these experiments are further explained in the preceding sections 3.1 and 3.2.

Minor Points:

1. Please increase the color contrast in graphs with multiple groups being plotted to make data more easily interpretable. Especially in Figures 1c, 2a, and 4b-e.

As suggested by the Reviewer, we adjusted the color contrast in multiple graphs, particularly ones containing the three categories SQLE sensitivities (sensitive, moderate, insensitive).

Reviewer #3: Expert in cholesterol metabolism

In this study, Mahoney and colleagues performed a drug screening to identify novel small molecules to target neuroendocrine tumors. The authors found that specific inhibitors for squalene epoxidase, the enzyme that produce the first sterol intermediate of cholesterol biosynthesis pathway suppress cellular proliferation and attenuates tumor growth in mice. Surprisingly, the authors demonstrate that the effects are mediated by the accumulation of squalene rather than the depletion of cholesterol. This finding is supported using other inhibitors of the cholesterol biosynthesis pathway including statins. While the study is of interest, the mechanism by which squalene influences cell survival and proliferation is not explored.

Major comments.

1) How and why accumulation of Squalene influences tumor cell survival and proliferation needs to be addressed. Is the plasma membrane composition affected?; How about oncogenic signaling, numerous oncogenes (RAS, etc) are located in the plasma membrane.

The Reviewer highlights the need for additional mechanistic studies to further understand the downstream consequences of SQLE inhibition. While the full understanding of different roles that squalene could play within a cell is beyond the scope of this initial manuscript, we have taken a

significant step in this direction by characterizing the role of lipid droplets as a squalene storage depot. This is likely to be a significant part of the overall mechanism involved as we can modulate the NB-598 response by modulating lipid droplet status. Please see our detailed response to Reviewer #2, points 3.1-3.3.

In the updated manuscript we readily acknowledge in the Discussion section that we still lack all the mechanistic details of cellular response to accumulating squalene that exceeds the lipid droplet storage capacity. The Reviewer correctly points out that squalene, given its extremely hydrophobic nature, is likely to have pleiotropic effects on multiple membranes and organelles. Future studies will focus on the dissection of these mechanisms and determining whether this subset of SCLC cell lines displays a broader defect in lipid handling capacity.

2) Tumor morphology should be characterized morphologically. Is the SQLE inhibitor affecting tumor cell apoptosis, proliferation, angiogenesis, tumor cell macrophages, etc. These studies are superficial.

The Reviewer suggests a number of studies to more deeply characterize the NB-598 treated tumors *in vivo*. We have conducted some of these analyses focusing primarily on apoptosis (using cleaved caspase 3 marker) and proliferation (using Ki-67 marker) and the data is now shown in **Supplementary Figure 6**. While we didn't find any significant differences between the vehicle-treated versus NB-598-treated tumors, there are several important caveats worth emphasizing:

- The H1963 tumors utilized for staining were from the end of the efficacy study, where there is already a dramatic difference in the tumor size (**Figure 3d**). A detailed time course experiment focusing on earlier time points would be more likely to identify differences in cellular proliferation or apoptosis. We make this point clearly in the updated manuscript.
- All the efficacy studies are conducted in the context of immunocompromised mice to enable the studies using human SCLC cell lines. While we recognize the need to analyze the contribution of the immune system to the tumor ecosystem, such studies will have to await the identification of mouse SCLC cell lines, or models, that are sensitive to SQLE inhibition.

3) The efficacy of SQLE inhibitors in regulating tumor growth in vivo should be compared to other cholesterol biosynthesis inhibitors (statins).

We thank the Reviewer for this suggestion. However, we'd like to point out that we have done a number of *in vitro* analyses demonstrating the uniqueness of NB-598 sensitivity as compared to other cholesterol pathway inhibitors (**Figure 5g**). These results argue that our observations are not related to general cholesterol depletion but rather are associated with particular sensitivity to accumulating squalene. In fact, co-treatment with pathway inhibitors upstream of SQLE, such as statins and lapaquistat, results in rescue of growth defects (**Figure 5e** and **Supplementary Figure 7c,d**).

Furthermore, these conclusions are strengthened by the newly included genetic double KO results, wherein growth defects mediated by sgRNAs targeting SQLE are rescued in the context of FDFT1 KO cells (**Figure 5d**). Collectively, our data argues that the cellular sensitivity we identified is unrelated to general cholesterol depletion.

Therefore, the Reviewer's suggestion, while interesting, is not fully aligned with the main point of the manuscript. Instead, we focused the bulk of our effort to address Reviewer's point #1 as deeply as possible.

Overall, the authors report an interesting finding but they should provide better characterization of the in vivo tumor model and identified the molecular mechanism by which SQLE mediates cytotoxic effects in neuroendocrine tumor cell lines.

We thank the Reviewer for the recognition of the novelty of our work.

Reviewers' Comments:

Reviewer #1:

Remarks to the Author:

The authors have done a very good job of addressing the comments regarding the previous submission. The manuscript is now suitable for publication.

Reviewer #2:

Remarks to the Author:

The authors have addressed most of this reviewer's concerns with the additional data included in the revised manuscript.

In cases where they performed studies to address comments (point 2), they provide substantial justification as to address the caveats of those experiments.

This manuscript is acceptable for publication in its current state.

Reviewer #3:

Remarks to the Author:

While the paper has improved, there are still many caveats of how the accumulation of squalene influences the proliferation of tumor cells. I disagree that this aspect is outside of the scope of this study. The accumulation of lipid droplets in cells treated with squalene epoxidase is not indicative of the accumulation of squalene in lipid droplets (lipid droplets stain for neutral lipids (sterol esters and TAGs)). The functional effect of lipid droplet accumulation and cellular proliferation is not studied.

The characterization of the tumors is merely descriptive and not qualified.

Manuscript NCOMMS-18-03313B

Response to reviewers.

REVIEWERS' COMMENTS:

Reviewer #1 (Remarks to the Author):

The authors have done a very good job of addressing the comments regarding the previous submission. The manuscript is now suitable for publication.

We thank the Reviewer for their time and their suggestions, which have greatly improved the manuscript.

Reviewer #2 (Remarks to the Author):

The authors have addressed most of this reviewers concerns with the additional data included in the revised manuscript.

In cases where they performed studies to address comments (point 2), they provide substantial justification as to address the caveats of those experiments.

This manuscript is acceptable for publication in its current state.

We thank the Reviewer for their time and their suggestions, which have greatly improved the manuscript.

Reviewer #3 (Remarks to the Author):

While the paper has improve. There are still many caveats of how the accumulation of squalene influences the proliferation of tumor cells. I disagree that this is aspect is outside of the scope of this study. The accumulation of lipid droplets in cells treated with squalene epoxidase is not indicative the accumulation of squalene in lipid droplets (lipid droplets stain for neutral lipids (sterol esters and TAGs). The functional effect of lipid droplet accumulation and cellular proliferation is not study. The characterization of the tumors is merely descriptive and not qualified.

We thank the Reviewer for their time and the recognition that the manuscript is significantly improved as compared to the original submission.

Furthermore, we agree with the Reviewer's assessment that we don't have the entire mechanism of squalene action elucidated in this manuscript. However, the current manuscript already provides multiple novel observations:

- Large-scale cell line panel screen identifying novel sensitivity to SQLE inhibition.

- Careful characterization of previously published SQLE inhibitor, NB-598.
- In vivo demonstration of SQLE sensitivity in multiple xenograft models.
- Demonstration that squalene accumulation is necessary for NB-598 effects in SQLE-sensitive cell lines.
- The role of lipid droplets as squalene storage depots.

Therefore, we respectfully would like to argue that the interesting studies suggested by the Reviewer are beyond the scope of this initial manuscript and will be the subject of future experiments.